# Structure and topography of the synaptic V-ATPase–synaptophysin complex

Chuchu Wang[1,2,3,4,5,12], Wenhong Jiang[6,12], Jeremy Leitz[1,2,3,4,5,12], Kailu Yang[1,2,3,4,5], Luis Esquivies[1,2,3,4,5], Xing Wang[6], Xiaotao Shen[7,8,9], Richard G. Held[1,2,3,4,5], Daniel J. Adams[10], Tamara Basta[10], Lucas Hampton[10], Ruiqi Jian[7,8], Lihua Jiang[7,8], Michael H. B. Stowell[10], Wolfgang Baumeister[11], Qiang Guo[6✉] & Axel T. Brunger[1,2,3,4,5✉]

Synaptic vesicles are organelles with a precisely defined protein and lipid composition[1,2], yet the molecular mechanisms for the biogenesis of synaptic vesicles are mainly unknown. Here we discovered a well-defined interface between the synaptic vesicle V-ATPase and synaptophysin by in situ cryo-electron tomography and single-particle cryo-electron microscopy of functional synaptic vesicles isolated from mouse brains[3]. The synaptic vesicle V-ATPase is an ATP-dependent proton pump that establishes the proton gradient across the synaptic vesicle, which in turn drives the uptake of neurotransmitters[4,5]. Synaptophysin[6] and its paralogues synaptoporin[7] and synaptogyrin[8] belong to a family of abundant synaptic vesicle proteins whose function is still unclear. We performed structural and functional studies of synaptophysin-knockout mice, confirming the identity of synaptophysin as an interaction partner with the V-ATPase. Although there is little change in the conformation of the V-ATPase upon interaction with synaptophysin, the presence of synaptophysin in synaptic vesicles profoundly affects the copy number of V-ATPases. This effect on the topography of synaptic vesicles suggests that synaptophysin assists in their biogenesis. In support of this model, we observed that synaptophysin-knockout mice exhibit severe seizure susceptibility, suggesting an imbalance of neurotransmitter release as a physiological consequence of the absence of synaptophysin.

Synaptic neurotransmission involves the fusion of neurotransmitter-filled synaptic vesicles with the presynaptic plasma membrane upon Ca[2+] influx into the presynaptic terminal. Synaptic vesicles are small organelles with an average diameter of approximately 40 nm and a specific composition of proteins and lipids[1,2]. The structure and function of some presynaptic proteins are known[9,10], along with a putative interaction map of synaptic vesicle proteins based on crosslinking mass spectrometry[11]. However, the molecular architecture of the whole synaptic vesicle is largely unknown, hindering the molecular mechanistic understanding of neurotransmitter release and its regulation.

## In situ maps of the V-ATPase

As a first step to decipher the molecular architecture of synaptic vesicles, we isolated and purified glutamatergic synaptic vesicles (ISVs) from mouse brains[1,2,12]. The ISVs are functional as assessed by a Ca[2+]-triggered vesicle–vesicle fusion assay[3]. We first imaged the ISVs with cryo-electron tomography (cryo-ET; Fig. 1a,b). Although cryo-ET can potentially resolve protein structures at near-atomic resolution by subtomogram averaging[13,14], the success depends on both the size of the molecule and the ability to localize it within a membrane environment. Here we focused on the synaptic vesicle vacuolar (H⁺)-ATPase (V-ATPase) as it has a large cytoplasmic domain readily observable in cryo-ET reconstructions (Fig. 1b).

V-ATPases constitute a highly conserved family of ATP-dependent proton pumps that are widely expressed in eukaryotic cells[15]. V-ATPases establish proton gradients critical for enabling organelle-specific functions, including membrane trafficking, endocytosis, lysosomal degradation and neurotransmitter release[5]. In addition to its role as a proton pump on synaptic vesicles, the synaptic vesicle V-ATPase has also been implicated in modulating neuronal exocytosis[4,16,17], although it is not directly involved in membrane fusion.

V-ATPases consist of an integral membrane V0 domain (also referred to as $V_o$) that functions as the proton pump and a cytosolic V1 domain (also referred to as $V_1$) that catalyses ATP hydrolysis. The V1 domain comprises three pairs of subunits responsible for ATP binding and hydrolysis, and its activity is coupled to the rotation of a central stalk in the V0 domain, mediating the translocation of protons across the

[1]Department of Molecular and Cellular Physiology, Stanford University, Stanford, CA, USA. [2]Department of Neurology and Neurological Sciences, Stanford University, Stanford, CA, USA. [3]Department of Structural Biology, Stanford University, Stanford, CA, USA. [4]Department of Photon Science, Stanford University, Stanford, CA, USA. [5]Howard Hughes Medical Institute, Stanford University, Stanford, CA, USA. [6]State Key Laboratory of Protein and Plant Gene Research, School of Life Sciences and Peking-Tsinghua Center for Life Sciences, Peking University, Beijing, China. [7]Department of Genetics, Stanford University, Stanford, CA, USA. [8]Stanford Center for Genomics and Personalized Medicine, Stanford University, Stanford, CA, USA. [9]Lee Kong Chian School of Medicine, Nanyang Technological University, Singapore, Singapore. [10]Department of Molecular, Cellular, and Developmental Biology, University of Colorado, Boulder, CO, USA. [11]Department of Structural Biology, Max Planck Institute of Biochemistry, Martinsried, Germany. [12]These authors contributed equally: Chuchu Wang, Wenhong Jiang, Jeremy Leitz. ✉e-mail: guo.qiang@pku.edu.cn; brunger@stanford.edu

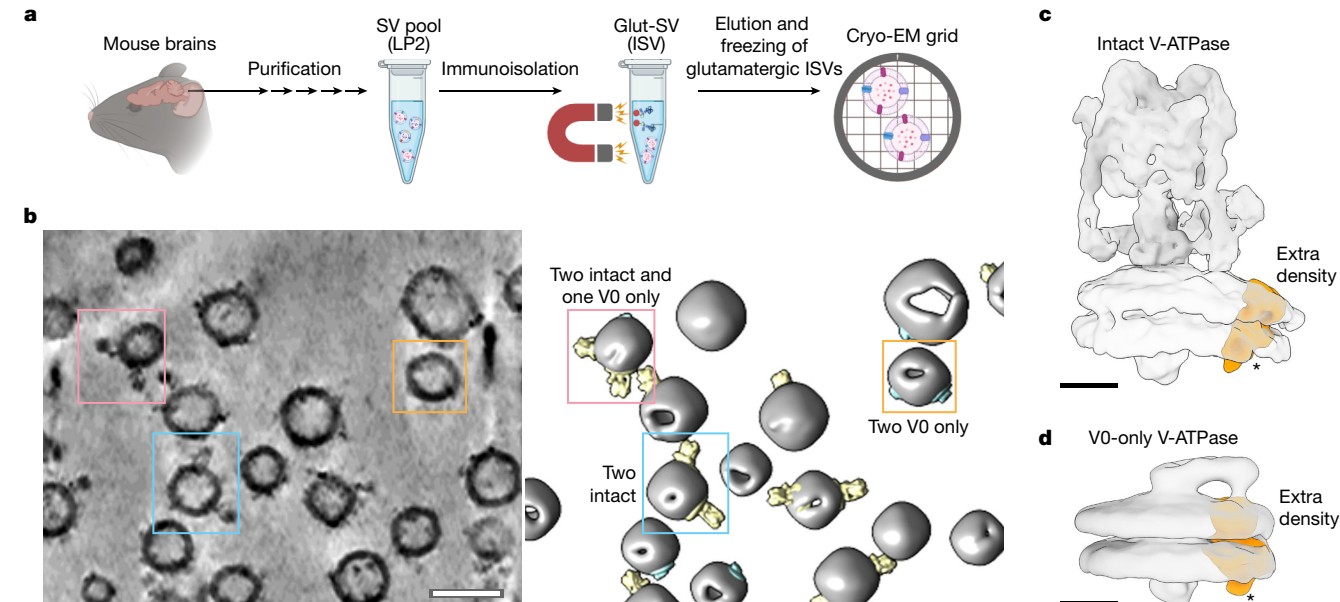

**Fig. 1 | Cryo-ET of glutamatergic ISVs. a**, Schematic overview of glutamatergic synaptic vesicle isolation and cryo-EM sample preparation (Methods). LP2, second lysis pellet. Schematic overview in **a** was created using BioRender (https://biorender.com). **b**, A representative tomogram (1 of 52 tomograms) (cryoCARE denoised) of ISVs is shown as a tomographic slice (left; thickness 1 nm) with a corresponding 3D rendering (right; Methods). The ISV membrane, intact and V0-only V-ATPase assemblies are coloured in grey, yellow and cyan, respectively. Three representative ISVs are boxed and annotated with the copy numbers of intact and V0-only V-ATPase assemblies. Scale bar, 50 nm. **c**,**d**, Subtomogram averaging maps (Methods and Extended Data Fig. 1) of intact (**c**; state 3) and V0-only (**d**) V-ATPase assemblies (transparent surfaces). The extra binding partner density (not part of the V-ATPase assembly) is coloured in orange and denoted with an asterisk. Scale bars, 5 nm.

membrane[18–23]. The V1 domain reversibly disassembles from the V0 domain in a luminal pH-dependent manner[4]. Several V-ATPase subunits have multiple isoforms with distinct tissue expression, suggesting tissue-specific roles[24].

In our cryo-ET reconstructions of ISVs, we found both 'intact' (that is, V0 and V1) and V0 domain-only V-ATPase assemblies in the ISVs (Fig. 1b), demonstrating that a mixture of intact and V0-only V-ATPases exists in functional ISVs, consistent with observations of V1 domain dissociation[4]. We then performed subtomogram averaging and classification (Fig. 1c,d and Extended Data Fig. 1), yielding three well-defined states of the intact V-ATPase at approximately 17 Å resolution corresponding to about 120° rotations of the rotor subcomplex between the states. Our data also produced one state for the V0-only assembly, consistent with previous work[25]. Both the intact and the V0-only V-ATPase maps revealed an extra density large enough to fit an approximately 20 kDa protein (Fig. 1c,d), which, to our knowledge, had not been observed in any structures of purified V-ATPases[18,22,23,25,26], suggesting that the extra density comprises at least one specific binding partner that exists only in the context of synaptic vesicle membranes.

## Identification of the binding partner

To achieve higher-resolution maps and resolve the binding partner better, we collected a large set of cryo-electron microscopy (cryo-EM) projection images and performed single-particle cryo-EM analysis (SPA) of the V-ATPase (Fig. 2a,b and Extended Data Fig. 2). Refinements resulted in maps of the intact V-ATPase assembly, again consisting of three states all at 4.3 Å resolution and a map of the V0-only V-ATPase assembly at 3.8 Å resolution. Note that the resolution of our V0-only map exceeds that of a previous mammalian V0-only V-ATPase structure[27] and is comparable with that of a recently reported V0 structure in nonspecifically isolated synaptic vesicles[28]. All of these maps clearly show the density of the binding partner.

To identify the binding partner, we generated density maps from models predicted by AlphaFold2[29] for all synaptic proteins detected in our ISVs by high-resolution mass spectrometry (Supplementary Table 1). The predicted density maps were fitted into the densities of the binding partner for the intact (state 3) and V0-only V-ATPase assemblies using CoLoRes/Situs[30,31]. The rigid-body fitted models were sorted by their map cross-correlation coefficient (Supplementary Table 1). Candidates that were not membrane proteins or had no membrane domain matching the binding partner density in the membrane region were ignored for further analysis. Models with the highest cross-correlation coefficient were checked and adjusted by inspecting the models and maps, and the model outlier percentage was calculated (Methods). Synaptophysin (also known as synaptophysin-1 (SYP1))[6] exhibited the least outliers, followed by its paralogues (Fig. 2c): synaptoporin (also known as synaptophysin-2)[7], synaptogyrin-1 (SYG1) and synaptogyrin-3[8]. We compared the Alpha-Fold2 prediction of SYG1 with the recently determined NMR structure of SYG1 (Protein Data Bank (PDB) ID 8A6M)[32] and found the experimental structure to be similar to the predicted model (root-mean-square difference = 1.4 Å).

Consistent with our fitting results, a crosslinking mass spectrometry study of ISVs[11] had identified a possible interaction between synaptophysin and the synaptic V-ATPase. However, this crosslinking study used crosslinkers that act on the cytoplasmic site of the ISV, potentially limiting the detection of luminal interactions. Moreover, further supporting our findings, the V0 domain of the synaptic vesicle V-ATPase was associated with synaptophysin in synaptosome preparations[33]. Together, synaptophysin is the most likely candidate for the binding partner to the synaptic vesicle V-ATPases that we discovered in our in situ cryo-ET and single-particle cryo-EM studies of ISVs.

## V-ATPase–synaptophysin validation

To further confirm the identity of the binding partner, we used synaptophysin-knockout ($Syp^{-/-}$) mice (Methods), and purified the $Syp^{-/-}$ ISVs following the same protocol as for wild-type mice. As anticipated, essentially no SYP1 was detected in the $Syp^{-/-}$ ISVs (Fig. 2d and

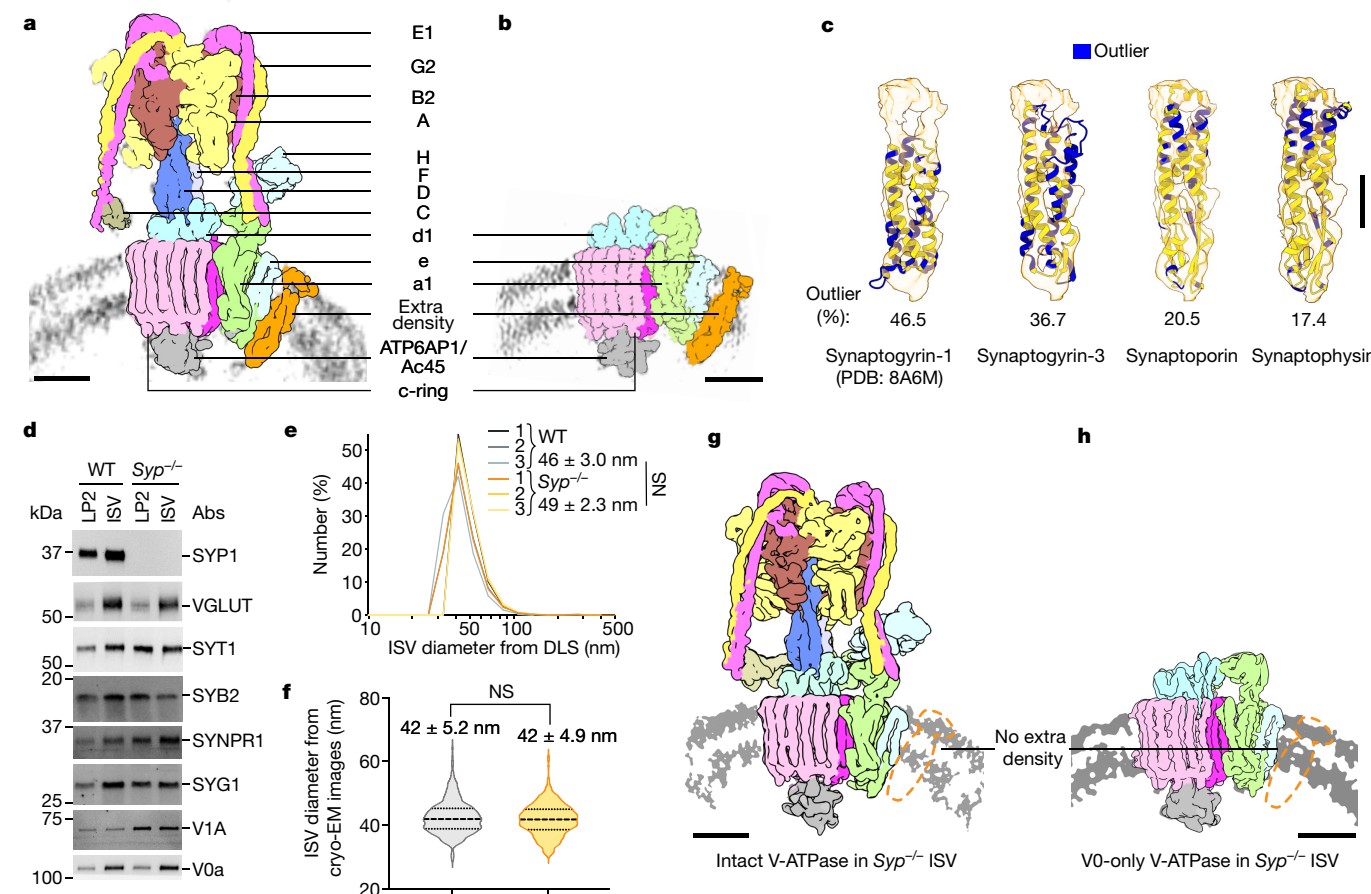

**Fig. 2 | Maps of V-ATPases imaged in wild-type and *Syp*[−/−] ISVs. a,b**, SPA maps of intact (**a**; state 3) and V0-only (**b**) V-ATPase assemblies imaged in wild-type (WT) ISVs. The V-ATPase subunits are coloured as indicated, and the extra binding partner density is coloured in orange. The background grey arc represents the ISV membrane density. Scale bars, 5 nm. **c**, Docking of models into the binding partner density (orange) extracted from the map shown in **a**. The solution NMR structure of synaptogyrin-1 and the AlphaFold2 predicted atomic models of synaptogyrin-3, synaptoporin and synaptophysin are shown. The atomic models are shown in cartoon representation and coloured in yellow and blue to indicate if the model is in the interior or the exterior of the map, respectively. For clarity, only the transmembrane and luminal parts are displayed, omitting the cytosolic regions that are predicted to be unstructured. Scale bar, 5 nm. **d**, Representative western blots (one of nine independent measurements) of synaptic proteins in LP2 and ISV samples from wild-type and *Syp*[−/−] ISVs (Extended Data Fig. 4). Abs, antibodies. **e**, Dynamic light scattering (DLS) measurements of wild-type and *Syp*[−/−] ISVs. The means and standard deviations of the ISV diameters were calculated from three independent ISV preparations. *P*-values were calculated by the unpaired two-tailed *t*-test. NS, not significant. **f**, Size distribution analysis of wild-type and *Syp*[−/−] ISVs by inspection of cryo-EM images. The mean ± s.d. of wild-type and *Syp*[−/−] ISV diameters were calculated from 326 wild-type ISVs and 362 *Syp*[−/−] ISVs, respectively. In the violin plots, the bottom dotted line represents the first quartile, the middle dashed line represents the median, and the top dotted line represents the third quartile. *P*-values were calculated by unpaired two-tailed *t*-test. **g,h**, SPA maps of intact (**g**; state 3) and V0-only (**h**) V-ATPase assemblies imaged in *Syp*[−/−] ISVs. The V-ATPase subunits are coloured as in **a** and **b**. The corresponding location of the binding partner density identified in wild-type ISVs is indicated as a dashed orange silhouette for comparison. The background grey arc represents the ISV membrane density. Scale bars, 5 nm.

Extended Data Fig. 3a,b) as determined by western blot. Vesicular glutamate transporter 1 (VGLUT1) and synaptotagmin-1 (SYT1) were present at similar levels in *Syp*[−/−] ISVs and wild-type ISVs (Fig. 2d and Extended Data Fig. 3c). Using VGLUT1 (Extended Data Fig. 3d) or SYT1 (Extended Data Fig. 3e) for band density normalization, synaptoporin-1 (SYNPR1) was present at a higher level, and SYG1 was present at a similar level in the *Syp*[−/−] ISVs. By contrast, synaptobrevin-2 (SYB2) was present at a lower level in *Syp*[−/−] ISVs than in the wild-type ISVs, consistent with a role of synaptophysin for SYB2 sorting into synaptic vesicles[34–36]. The observed V-ATPase level, either targeting subunit A (V1A) or subunit a (V0a) of the V-ATPase, was higher in *Syp*[−/−] ISVs.

The hydrodynamic size distributions of wild-type and *Syp*[−/−] ISVs were very similar in solution as determined by dynamic light scattering (Fig. 2e), consistent with size measurements by cryo-EM (Fig. 2f and representative images in Extended Data Figs. 2a and 4a). Finally, the binding partner density was absent in the *Syp*[−/−] ISV SPA maps for

both intact and V0-only V-ATPase assemblies (Fig. 2g,h and Extended Data Fig. 4b–f), validating the identity of the binding partner density as synaptophysin.

## V-ATPase–synaptophysin structure

Starting with published V-ATPase structures and the AlphaFold2-predicted synaptophysin model, we refined atomic models of both the intact and the V0-only V-ATPase assemblies in complex with synaptophysin using the wild-type ISV SPA average maps (Methods, Figs. 2a,b and 3a,b and Extended Data Table 2). Generally, there is good agreement between the models and maps (Extended Data Fig. 5a–c). We observed the glycosylated mammalian-specific ATP6AP1-binding partner in our maps (Extended Data Fig. 5d). In addition to six previously reported glycosylation sites[26], a putative glycosylation site was also observed for Asn399.

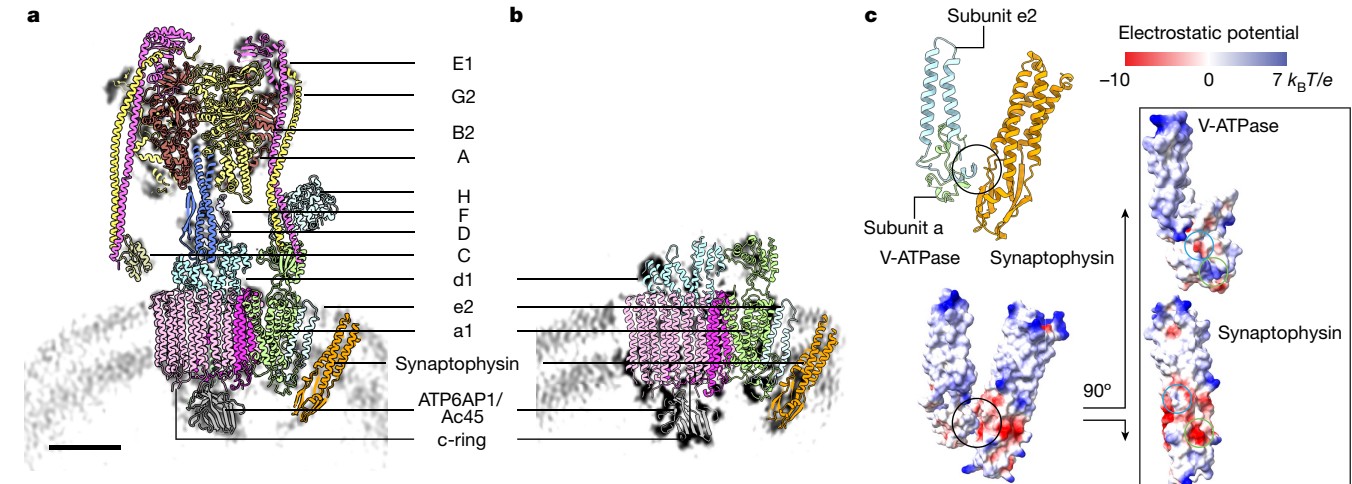

**Fig. 3 | Structures of V-ATPases imaged in wild-type ISVs. a,b**, Atomic models of the V-ATPase–synaptophysin complex imaged in wild-type ISVs. The intact (**a**; state 3) and V0-only (**b**) V-ATPase assemblies are shown. The V-ATPase subunits are coloured as indicated, and synaptophysin is coloured in orange. The background grey arc represents the ISV membrane density. Scale bar, 5 nm (the same scale bar is used in **a** and **b**). **c**, Interface (black circle) between V-ATPase subunits e2, a and synaptophysin (the structure of the V0-only V-ATPase assembly of wild-type ISVs). The electrostatic surface potential of this interface is also shown along with an 'open book' view. The blue and green circles highlight the interface between synaptophysin and V-ATPase subunit e2 and a, respectively. The interface area is approximately 350 Å².

Although both e1 (ATP6V0E1) and e2 (ATP6V0E2) isoforms have been reported for subunit e, our high-resolution V0-only V-ATPase map suggests e2 as the subunit present in our sample (Extended Data Fig. 6a). This finding was further supported by mass spectrometry experiments, which only detected unique peptides specific to ATP6V0E2 (Supplementary Table 2). The luminal parts of subunit e2 and subunit a form an interface with synaptophysin (Fig. 3c), and electrostatic interactions are involved in this interface.

Synaptoporin shares function and primary sequence similarities with synaptophysin, and it is present at a slightly higher level in *Syp*⁻/⁻ ISVs (Extended Data Fig. 4c–e), potentially to compensate for the loss of synaptophysin function. However, it is improbable that it interacts quantitatively with the V-ATPase as there is no density for it in the *Syp*⁻/⁻ ISV SPA maps (Fig. 2g,h). The absence of an interaction probably arises owing to primary sequence differences between synaptophysin and synaptoporin at and near the V-ATPase–synaptophysin interface (Extended Data Fig. 6b–d), suggesting the specificity of this interaction.

At the current resolution, we do not observe substantial differences in maps and models between the V-ATPases in wild-type and *Syp*⁻/⁻ ISVs (Extended Data Fig. 7). Overall, our V-ATPase models are similar to structures of mammalian V-ATPases from purified samples (PDB IDs 6WM3, 7U4T, 6VQG and 7UNF[18,23,26]; Supplementary Table 3) and recently reported structures from nonspecifically isolated synaptic vesicles[28,37].

## *Syp*⁻/⁻ increases V-ATPase copy numbers

Considering that the interaction between synaptophysin and the V-ATPase does not appear to affect the V-ATPase structure, we asked what other roles this interaction might have. We collected a cryo-ET dataset (Fig. 4a and Extended Data Fig. 8a) for *Syp*⁻/⁻ ISVs using the same procedures as for the cryo-ET dataset for wild-type ISVs (Extended Data Fig. 8b). We inspected the cryo-ET reconstructions and identified intact and V0-only V-ATPase assemblies in both wild-type and *Syp*⁻/⁻ ISV datasets, which allowed us to analyse the V-ATPase copy numbers (Methods and Extended Data Fig. 8a,b). There is little correlation between the V-ATPase copy numbers and ISV diameters for both wild-type and *Syp*⁻/⁻ ISVs (Extended Data Fig. 8c).

Compared with wild-type ISVs, we observed a substantial increase in the average copy number of V-ATPases in *Syp*⁻/⁻ ISVs (Fig. 4b,c, Extended Data Fig. 8d,e and Supplementary Tables 4 and 5). Not only were there more ISVs with more than two V-ATPases but we also observed ISVs with up to eight V-ATPases on a single ISV. To estimate the average copy number of V-ATPases per ISV, we fitted Poisson distributions to the observed copy numbers (Extended Data Fig. 8f–i). However, as it is difficult to determine the true copy number of ISVs without any V-ATPase assembly owing to the missing wedge effect of tomographic reconstructions, we only fitted the Poisson distributions to copy numbers of 1 or more. The Poisson distributions could be well fit to the copy numbers of 1 or more (Extended Data Fig. 8f–i), suggesting that the incorporation of a V-ATPase is a Poisson process, that is, independent of the presence of other V-ATPases in the same synaptic vesicle. Therefore, we used the λ parameters of the fitted Poisson distributions to estimate the average copy numbers. For *Syp*⁻/⁻ ISVs, there is a 2.1-fold increase in the average copy number of intact V-ATPase assemblies and a 1.7-fold increase in the average copy number of V0-only assemblies.

For wild-type ISVs, the majority (71.3% of wild-type ISVs with visible V-ATPase densities) contained only one intact V-ATPase assembly (Fig. 4b and Supplementary Table 4). The remaining ISVs (21.9%) primarily contained two intact V-ATPase assemblies, along with a few cases (6.8%) with more than two copies. We calculated the average combined copy number of intact and V0-only V-ATPases from the Poisson fits to be 1.04 (sum of 0.62 and 0.42, respectively). These numbers are in close agreement with the previously estimated copy number of 0.7 intact V-ATPases per ISV using quantitative western blotting against the V1B subunit of the V-ATPase[1] and an estimate of the combined copy number of intact and V0-only V-ATPases per ISV of 1.27 using fluorescent imaging with antibody labelling[38].

## *Syp*⁻/⁻ causes severe seizures in mice

Given the profound dysregulation of V-ATPase copy number observed in *Syp*⁻/⁻ ISVs, we examined what physiological consequences this may have. Although *Syp*⁻/⁻ mice exhibit no large differences in neurotransmitter release probability compared with wild-type mice[39,40], we observed a striking susceptibility to kainic acid stress-induced seizures (Fig. 4d). Compared with wild-type controls displaying minimal susceptibility, *Syp*⁻/⁻ mice are highly susceptible to seizure and subsequent death. This dramatic phenotype highlights a significant role for synaptophysin in synaptic function and in the regulation of synaptic vesicle

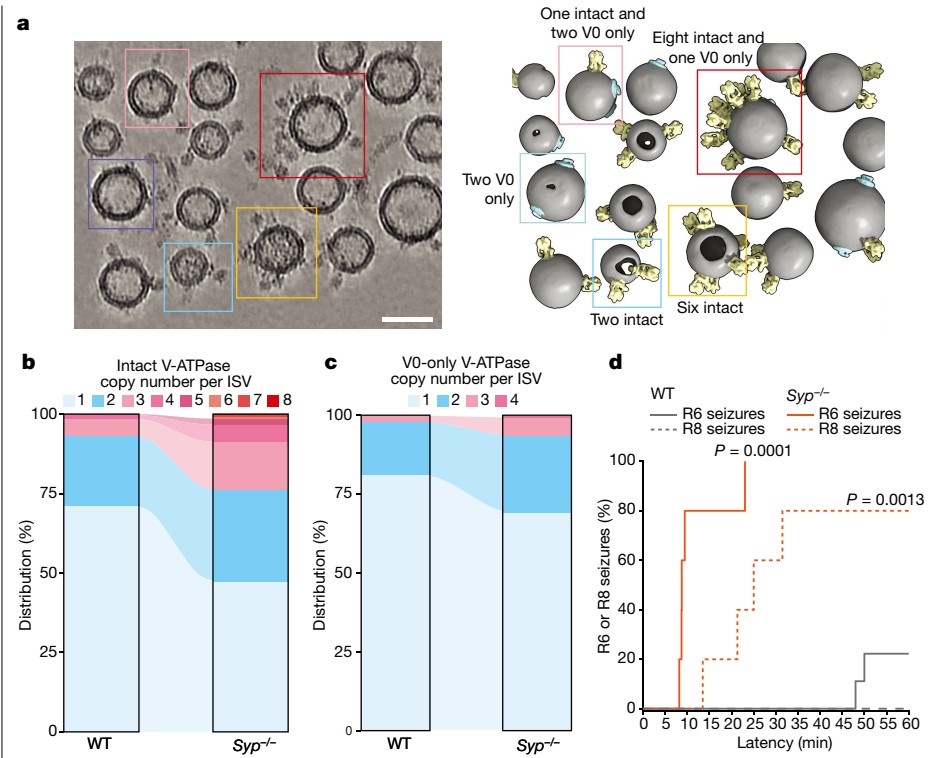

**a**

One intact and
two V0 only

Eight intact and
one V0 only

Two V0
only

Two intact    Six intact

**b** Intact V-ATPase
copy number per ISV

**c** V0-only V-ATPase
copy number per ISV

**d**

**Fig. 4 | Increase of V-ATPase copy numbers and seizures in *Syp*⁻/⁻ mice. a**, A representative tomogram (1 of 78 tomograms; cryoCARE denoised) of ISVs from *Syp*⁻/⁻ mouse brains is shown as a tomographic slice (left; thickness of 35.5 nm) with a corresponding 3D rendering (right). The ISV membrane, and the intact and V0-only V-ATPase assemblies are coloured in grey, yellow and cyan, respectively. Five representative ISVs are boxed and labelled, with the copy number of intact and V0-only V-ATPase assemblies identified within each. More examples and views are shown in Extended Data Fig. 8a,b. Scale bar, 50 nm. **b,c**, Distribution of intact (**b**) and V0-only (**c**) V-ATPase copy number per wild-type ISV or *Syp*⁻/⁻ ISV. Statistical significance tests are shown in Extended Data Fig. 8d,e, and numerical data are available in Supplementary Tables 4 and 5. **d**, Mice of 4–6 months of age of two genotypes (wild-type (*n* = 9) and *Syp*⁻/⁻ (*n* = 5)) were injected with kainic acid (25 mg kg⁻¹ intraperitoneally) and observed for 60 min with video recording. Seizure severity was scored blind by two observers using a modified Racine scale, and the latency survival to R6 or R8 seizure severity is reported (Methods). **\*\****P* = 0.0013 and **\*\*\****P* < 0.0001 (one-sided Gehan–Breslow–Wilcoxon test).

properties. Although there may be other roles of synaptophysin at the synapse, the severe seizure susceptibility phenotype and the regulation of V-ATPase copy number highlight a previously unrecognized and critical role for synaptophysin in synaptic vesicle function.

## Discussion

We performed in situ cryo-ET reconstructions, obtained SPA maps and determined structures of synaptic vesicle V-ATPases in functional, glutamatergic mouse ISVs. We observed both intact and V0-only V-ATPase assemblies (Figs. 1b–d and 2a,b), consistent with the notion that V-ATPases reversibly dissociate the V1 domain upon acidification of the synaptic vesicle[4]. Unexpectedly, the cryo-ET reconstructions and SPA maps revealed that synaptophysin is bound to the synaptic vesicle V-ATPase. This interaction was not previously observed in structural studies of reconstituted V-ATPases[18,22,23,25,26,41], illustrating the power of in situ structural studies by cryo-ET and single-particle cryo-EM. The confined membrane environment of a synaptic vesicle, including vesicle curvature, probably contributes to stabilizing the relatively small V-ATPase–synaptophysin interface (Fig. 3c). The interaction is identical for both intact and V0-only V-ATPase assemblies (Fig. 2a,b), suggesting that this interaction does not appear to directly regulate the hydrolysis mechanism of the V-ATPase.

SPA maps of *Syp*⁻/⁻ ISVs validated the V-ATPase–synaptophysin interaction and demonstrated the specificity of this interaction for synaptophysin, but not for the paralogues synaptoporin and synaptogyrin (Fig. 2d,g,h and Extended Data Fig. 6c,d). Comparison of the structures shows that the V-ATPase structure is not substantially affected by the interaction with synaptophysin (Extended Data Fig. 7 and Supplementary Table 3).

At the gross morphological level, *Syp*⁻/⁻ mice form normal synapses and synaptic vesicles[42], as corroborated by our cryo-ET reconstructions of *Syp*⁻/⁻ ISVs (Fig. 4a). *Syp*⁻/⁻ mice exhibit normal excitatory postsynaptic current release probability and miniature excitatory postsynaptic current frequency, although a small but significant increase in excitatory postsynaptic current quantal amplitude was previously observed[40]. More profound changes were observed with double genetic

deletion of both synaptophysin and synaptogyrin, resulting in reduced short-term and long-term plasticity in neurons[39]. Quadruple knockout studies produced an even larger increase in quantal amplitude[43]. We observed that *Syp*⁻/⁻ mice exhibit a severe seizure phenotype under pharmacological treatment with a glutamate agonist (Fig. 4d), which is concordant with the increase in quantal size observed for both *Syp*⁻/⁻ and quadruple knockout[40,43].

In addition to the interaction with the V-ATPase, synaptophysin also interacts with synaptobrevin[44–46]. Synaptophysin is a highly abundant synaptic vesicle protein with an estimated 30 copies in synaptic vesicles[1]. Consistent with the lower amount of synaptobrevin in *Syp*⁻/⁻ ISVs (Fig. 2d), synaptophysin has a role in synaptobrevin sorting into synaptic vesicles[34–36] and endocytosis[47], and presumably this sorting mechanism is assisted by the synaptophysin–synaptobrevin interactions. Despite the lower synaptobrevin amount in the *Syp*⁻/⁻ ISVs, the transmitter release probability is normal[40], consistent with the observation that two synaptobrevin molecules are sufficient for evoked synaptic vesicle fusion[48] compared with the much larger copy number of synaptobrevin molecules in wild-type ISVs (approximately 70 per vesicle)[1].

The copy number of V-ATPases roughly doubled in *Syp*⁻/⁻ ISVs (Fig. 4b,c and Extended Data Fig. 8d–i). A possible explanation is that the V-ATPase–synaptophysin interaction effectively increases the lateral membrane cross-section of the combined molecular assembly in wild-type ISVs. This increased membrane cross-section will limit the available area in the synaptic vesicle membrane owing to molecular crowding by increasing the excluded surface area[49], resulting in a lower V-ATPase copy number in wild-type ISVs than in *Syp*⁻/⁻ ISVs. Moreover, synaptophysin can form hexameric assemblies[50] and interact with synaptobrevin[11,46]. As we did not observe densities for synaptobrevin or synaptophysin oligomers in our cryo-ET and SPA maps (Figs. 1c,d and 2a,b), it is possible that these interactions, if they exist, are heterogeneous and averaged out in the cryo-EM maps. Nevertheless, these additional interactions would further increase the membrane cross-section of the combined assembly.

The increased V-ATPase copy number in *Syp*⁻/⁻ ISVs offers a possible explanation for the observed seizure phenotype under pharmacological treatment with a glutamate agonist (Fig. 4d), which is concordant

with the increase in the quantal size observed in $Syp^{-/-}$ studies[40,43]. This increased quantal size could potentially be different in excitatory and inhibitory synapses, as suggested by different synaptophysin copy numbers in glutamatergic and GABAergic wild-type ISVs[2]. Moreover, the increased copy number could exacerbate the leakiness of ISVs owing to ultraslow mode switching[51], further affecting this imbalance. Multiple studies have described seizure phenotypes due to mutations in the V-ATPase[52–55], and some of which are close to the V-ATPase–synaptophysin interface. The phenotypic similarity between these mutations and synaptophysin knockout, in conjunction with our data, suggests that the increased seizure susceptibility has a similar aetiology in both cases. Our observation of a substantially altered V-ATPase copy number per ISV is a plausible contributor to the observed phenotype. In summary, we showed that synaptophysin is a factor that assists in establishing the proper copy number of V-ATPases in synaptic vesicles, and loss of synaptophysin causes a severe stress-induced seizure phenotype probably due to an imbalance in neurotransmitter uptake and release.

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

## Methods

### Wild-type and *Syp*[−/−] mice

Male wild-type CD1 mice (23–26 days of age, purchased from Charles River) and C57BL/6 mice (23–26 days of age, purchased from Jackson Laboratory) were used for synaptic vesicle preparations. *Syp*[−/−] mice[42] were a gift from R. Leube. Male *Syp*[−/−] mice (23–26 days of age) were used for synaptic vesicle preparations. Both sexes of 4–6 months of age wild-type Black 6 (B6NTac; purchased from Jackson Laboratory) and *Syp*[−/−] mice were used in animal behavioural experiments. Mice were maintained in a colony room with a 12 h light–dark cycle (lights on at 07:00) with ad libitum access to food and water.

### *Syp*[−/−] mice genotyping

A 1-mm mouse ear notch was placed into a 1.5-ml microcentrifuge tube and treated with 200 µl of 50 mmol NaOH. Samples were heated to 95 °C in a heating block for 10 min. The tubes were cooled on ice for 10 min, and then 20 µl of Tris-HCl (pH 8.0) was added to neutralize the sample. The samples were centrifuged to pellet the tissue debris and the supernatant was used immediately for PCR using the following primers: 5′-ACTTCCATCCCTATTTCCCACACC-3′, 5′-TTCC ACCCACCAGTTCAGTAGGA-3′ and 5′-TCGCCTTCTTGACGAGTTCTT CTG-3′. The following cycle parameters were used: 3 min at 95 °C, 35 cycles of (30 s at 95 °C, 30 s at 58 °C and 40 s at 72 °C), 3 min at 72 °C, and hold at 4 °C. PCR products were analysed by agarose gel electrophoresis using a 1-kb ladder (N3231S, NEB). The wild-type WT fragment appeared at approximately 280 bp (256 expected), and the *Syp*[−/−] mutant fragment at approximately 650 bp (500 expected).

### Isolation of glutamatergic synaptic vesicles

Wild-type and *Syp*[−/−] glutamatergic ISVs were prepared and purified as previously described[1,2,12]. Eight wild-type mice (CD1 for wild-type TOMO data 1 only, specified below; all the other wild-type data are from C57BL/6 mice) and two *Syp*[−/−] mice were killed to purify the respective ISVs. Mouse brains were pooled in homogenizing buffer (50 ml 4 mM Na-HEPES (pH 7.4) and 320 mM sucrose; 9 ml for one brain) with protease inhibitors and homogenized in a 40-ml Dounce homogenizer four times with the 'A' pestle and three times with the 'B' pestle. The homogenate was centrifuged at 880*g* for 10 min. Supernatant (S1) was collected and centrifuged at 12,074*g* for 15 min. For the wild-type P2 pellet, the supernatant (S2) was removed, the P2 pellet was resuspended with 25 ml homogenizing buffer with protease inhibitors, the dark centre of the pellet was discarded, and the resuspension was transferred to a new tube and centrifuged at 27,167*g* for 15 min. However, the centrifugation step was skipped for the *Syp*[−/−] P2 pellet as the P2 sample size was small, following another previously published protocol[56]. The resulting wild-type pellet (P3) and *Syp*[−/−] P2, both of which contain synaptosomes, were resuspended with 5 ml or 1 ml homogenizing buffer and then hypo-osmotically lysed by 40 ml or 9 ml H$_2$O, respectively, followed by adding 1 M Na-HEPES (pH 7.4) and protease inhibitors to achieve a final concentration of 5 mM Na-HEPES. This solution was homogenized in a 40-ml or 25-ml Dounce homogenizer three times and then centrifuged at 43,589*g* for 20 min. The supernatant (LS1) was ultra-centrifuged at 256,631*g* for 2 h. The resulting pellet (LP2) containing different types of synaptic vesicles was then homogenized in a 2-ml Dounce homogenizer in 2 ml vesicle buffer (20 mM Na-HEPES (pH 7.4) and 90 mM NaCl). The homogenized synaptic vesicles were further mechanically sheered through a 27-gauge needle. Synaptic vesicle concentration was determined using a bicinchoninic acid assay, aliquoted, flash frozen in liquid nitrogen and stored at −80 °C until use.

To isolate the glutamatergic synaptic vesicles, 500 µg of LP2 fractions was thawed on ice and diluted by the blocking buffer (vesicle buffer + 0.5% (w/v) BSA) with 5 µg mouse VGLUT1 monoclonal antibody (135311, SySy) to achieve a final volume of 1 ml. This solution was incubated overnight at 4 °C and then added to 50 µl Dynabeads (10004D, Thermo Fisher Scientific); the pellet/sediment was incubated for another 2 h at 4 °C. Dynabeads were washed once in the blocking buffer and twice in vesicle buffer alone. Immuno-enriched VGLUT1-bound synaptic vesicles were then eluted three times by adding 25 µl of 40 mg ml[−1] (for the wild-type TOMO data 1 only, see below) or 1.1 mg ml[−1] (for all the other wild-type and *Syp*[−/−] ISV tomography and SPA data collections) VGLUT1 epitope peptide[12]. We tested the function of similar preparations of ISVs by a single-vesicle Ca[2+]-triggered fusion assay[3]. The size and concentration of freshly prepared ISVs were determined via dynamic light scattering (DLS) using a DynaPro NanoStar (Wyatt Technologies).

### SDS–PAGE and western blotting

The wild-type LP2, wild-type ISV, *Syp*[−/−] LP2 and *Syp*[−/−] ISV samples were diluted with Laemmli sample buffer (1610737, Bio-Rad) containing β-mercaptoethanol. The samples were loaded in TGX gels (4569036, Bio-Rad) and separated by electrophoresis with a standard sodium dodecyl sulfate (SDS) running buffer. For western blot analysis, samples in the gel were transferred to PVDF membranes by iBlot1 (Invitrogen). To verify the SYP depletion in the *Syp*[−/−] samples, the membrane was incubated with primary anti-rabbit synaptophysin-1 (1:1,000; 101008, SySy) and secondary goat anti-rabbit IRDye800CW (1:3,000; 926-32211, LI-COR) for fluorescent band detection by iBright1500 (Invitrogen). To detect the other synaptic proteins on this membrane for a fair comparison, the secondary antibody was stripped by Restore fluorescent western blot stripping buffer (62300, Thermo Scientific) and recovered by the EveryBlot blocking buffer (12010020, Bio-Rad). The primary and secondary antibodies were stripped from the western blot membranes following the manufacturer's instructions. We incubated the probed blot in the 1X stripping buffer for 15 min and washed it three times for 5 min each in TBST washing buffer, then incubated the membrane in EveryBlot blocking buffer for 5 min until reprobing. The membrane was incubated with primary anti-mouse SYB2 (1:1,000; 104211, SySy) and secondary goat anti-mouse horseradish peroxidase (HRP; 1:10,000; ab6789, Abcam) for chemiluminescent band detection by iBright1500 (Invitrogen). For one replicate (Extended Data Fig. 4a), the stripped membrane was cut at 37 kDa, and the upper membrane was incubated with primary anti-mouse SYT1 (1:1,000; 105011, SySy) and secondary goat anti-mouse HRP (1:10,000; ab6789, Abcam) for chemiluminescent band detection by iBright1500 (Invitrogen).

To compare different synaptic protein expression levels within a western blot, the membrane was cut into slices corresponding to 20 kDa, approximately 45 kDa, and 100-kDa ladder positions. The less than 20-kDa membrane slice was incubated with primary anti-mouse SYB2 (1:1,000) and secondary goat anti-mouse IRDye800CW (1:3,000; 926-32210, LI-COR). The 20–45-kDa membrane slice was incubated with primary anti-rabbit SYNPR1 (1:500; 102002, SySy) and secondary goat anti-rabbit IRDye800CW (1:3,000). The 45–100-kDa membrane slice was incubated with primary anti-rabbit ATP6V1A (1:500; 39517, CST) and secondary goat anti-rabbit IRDye800CW (1:3,000). The more than 100-kDa membrane slice was incubated with primary anti-rabbit ATP6V0A1 (1:1,000; NBP1-89342, NovusBio) and secondary goat anti-rabbit IRDye800CW (1:3,000). The fluorescent bands in the membrane slices were imaged by an iBright1500 imaging system (Thermo Fisher Scientific). To analyse other synaptic proteins on the same membrane, the secondary antibodies of the 20–45-kDa and 45–100-kDa membrane slices were then stripped and recovered. The primary and secondary antibodies were stripped from the western blot membranes following the manufacturer's instructions (62300, Thermo Fisher). We incubated the probed blot in the 1X stripping buffer for 15 min and washed it three times for 5 min each in TBST washing buffer, then incubated the membrane in 5% BSA blocking buffer for 30 min until reprobing. The 20–45-kDa membrane slice was incubated with primary anti-rabbit SYG1 (1:500; 103002, SySy) and secondary goat anti-rabbit IRDye800CW (1:3,000), and fluorescent bands were imaged by the

iBright1500 imaging system. The stripped 45–100-kDa membrane slice was incubated with a mixture of primary anti-rabbit VGLUT1 (1:1,000; 135303, SySy) and anti-mouse SYT1 (1:1,000). The fluorescent band of VGLUT1 was imaged after incubation of the secondary goat anti-rabbit IRDye800CW (1:3,000); the chemiluminescent band of SYT1 was imaged after incubation of the secondary goat anti-mouse HRP (1:10,000).

Densitometry and normalization were performed with the iBright analysis software (Thermo Fisher Scientific).

## Cryo-EM sample preparation

The freshly isolated wild-type ISVs or *Syp*[−/−] ISVs were mixed with or without pellets of the AURION 10-nm BSA Gold Tracer (25487, EMS) solution (v/v = 5/2) for the cryo-EM grids preparation. For the wild-type samples, Quantifoil (R2/4, 300 mesh) gold grids with a continuous layer of graphene (Graphenea) were plasma cleaned for 30–40 s by using a Fischione Nanoclean 1070 plasma cleaner (30–34 W forward power, «2 W reverse power and hydrogen delivered at 20 sccm). For the *Syp*[−/−] ISV samples, Quantifoil (R2/1, 300 mesh) gold holey-carbon grids (Q3100AR1, EMS) were used without pre-treatment. Grids were then placed on parafilm in a custom humidity chamber with approximately 2 ml of MilliQ water added to the periphery of the chamber. Of ISVs, 3.5–4 µl were applied to the grids and incubated for 8–15 min. Grids were then directly loaded onto a Vitrobot Mark I (FEI) and an additional 3 µl of ISVs were applied. Grids were then back-blotted by replacing the sample side blot paper with a custom-cut Teflon sheet or parafilm with a blot force of 5, a wait time of 1–10 s at 22 °C and 95% relative humidity. For each round of freezing, we froze grids with varying blotting times of 1–4 s. Grids were plunged into pure ethane and stored in liquid nitrogen until use.

## Tomographic data collection and reconstruction

For the first cryo-ET dataset of wild-type ISVs, grids were loaded onto a Titan Krios transmission cryo-electron microscopy (TEM) at the Max Planck Institute of Biochemistry operated at 300 kV and equipped with a field emission gun and a Gatan post-column energy filter. Images were recorded by a Gatan K2 Summit direct detector using dose fractionation mode (6.09 e$^−$ px$^{−1}$ s$^{−1}$ dose rate, 0.33 s exposure time per frame). Tilt series were collected using SerialEM[57] at a nominal magnification of ×53,000, resulting in a pixel size of 2.6 Å per pixel, with a defocus range between 4.0 and 5.0 µm. Of tomographic tilt series, 74 were collected using a dose-symmetric tilt scheme[58] between ±60° with an increment of 3°. The same exposure time (3.3 s exposure time and 10 frames per image) was used for each tilt angle, resulting in a total dose of around 120 e$^−$ Å$^{−2}$ per tilt series. We refer to this dataset as wild-type TOMO data 1 in Extended Data Table 1, and it was used for the experiments shown in Fig. 1b–d and Extended Data Fig. 1. Movie frames were aligned using Motioncor2[59], and motion-corrected tilt series were then aligned using fiducial tracking with the IMOD software package[60].

A second cryo-ET dataset of wild-type ISVs (used in Fig. 4b,c and Extended Data Fig. 8b, we refer to this dataset as wild-type TOMO data 2 in Extended Data Table 1) and the cryo-ET dataset of *Syp*[−/−] ISVs (used in Fig. 4a–c and Extended Data Fig. 8a, we refer to this dataset as *Syp*[−/−] TOMO data 1 in Extended Data Table 1) were collected on a Titan Krios TEM at the Stanford Cryo-Electron Microscopy Center (cEMc) operated at 300 kV and equipped with a field emission gun and a Gatan post-column energy filter. Images of wild-type ISVs and *Syp*[−/−] ISVs were recorded by a Gatan K3 Summit direct detector using dose fractionation mode with SerialEM[57] using the PACEtomo script and a built-in dose-symmetric tilt scheme, respectively. Wild-type ISV (*n* = 203) and *Syp*[−/−] ISV (*n* = 109) tomographic tilt series data were collected using SerialEM in low-dose mode at a physical pixel size of 1.1 Å per pixel. The energy filter slit-width was set to 20 eV. Tilt series were collected using a dose-symmetric tilt scheme[58] between ±60° with an increment of 3°. The defocus was set to be 1.0–4.0 µm with a step of 0.3 µm per tilt. The dose per tilt was 3.4 e$^−$ Å$^{−2}$ with a per-frame dose of 0.34 e$^−$ Å$^{−2}$,

resulting in a total dose of 139.4 e$^−$ Å$^{−2}$ over 41 tilts. Movie frames were aligned using Motioncor2[59], and the motion-corrected tilt series were then aligned using the patch-based local motion alignment with the AreTomo software package[61].

For all three datasets, the tomograms were reconstructed from aligned tilt series using the weighted back projection approach in the IMOD software package[60]. After manual inspection, a subset of 52 tomograms from the wild-type TOMO data 1 dataset, 156 tomograms from the wild-type TOMO data 2 dataset and 78 tomograms from the *Syp*[−/−] TOMO data 1 dataset were kept for further analysis.

## Subtomogram averaging

For the analysis of wild-type TOMO data 1 experiments shown in Fig. 1 and Extended Data Fig. 1, the MATLAB (Mathworks) TOM toolbox[62] was used as a general platform for image processing. All the tomograms were binned (binning factor = 4, 10.4 Å per pixel) for processing. Six were used to generate the initial model. The template-free detection method PySeg[63] was used to identify membrane-bound complexes. In brief, all the synaptic vesicle membranes were segmented out by a template-matching approach, using hollow spheres with different diameters as templates. A discrete Morse theory-based algorithm was used to trace all the densities associated with synaptic vesicles. Subtomogram candidates were selected based on the direction and size of these densities. For each subtomogram, the normal vector of the attaching membrane was determined, and a rotational average was generated around this direction. The averages were classified by the affinity propagation method[63]. Classes with clear membrane attaching densities were combined for constrained alignment and classification using RELION (v2)[64]. The resulting structure clearly showed a V-ATPase (Extended Data Fig. 1a).

The resulting structure was then used as a new template to perform a new round of template matching using PyTom[65], but this time using all the binned tomograms. The resulting hits were manually sorted to produce the final dataset. In total, 1,860 V-ATPases were identified from 52 tomograms. Subsequently, the unbinned (full-size) tomograms were used for classification and averaging. Subtomograms of 192 × 192 × 192 pixel volume were extracted with Warp[66], with which 3D contrast transfer function (CTF) models were generated for each subtomogram. The classification was performed using the 3D tomography workflow of RELION (v2)[64], with a soft mask applied in the cytosolic domain. Three different conformations of the cytosolic domain were identified. A soft mask, including the entire molecular assembly, was used for the final refinement. Another round of multi-particle refinement was applied to the final maps using Warp-M[67], further refining the tilt series alignment parameters. For each subtomogram, only the first 15 subtilts (45 e$^−$ Å$^{−2}$ accumulative dose) were included for the final reconstruction and resolution estimation. The resolution was determined using the 0.143 criteria according to the gold-standard Fourier shell correlation[68].

## Tomogram segmentation and 3D rendering

To visualize the ISV membranes, we first used an automated tracing method based on tensor voting[69]. Subsequently, any necessary manual corrections were made using Amira software (Thermo Fisher Scientific). V-ATPase assemblies were positioned in their native locations and orientations. This positioning utilized low-pass-filtered reconstructed maps and the Euler angle information obtained from either subtomogram averaging (Fig. 1b) or template matching (Fig. 4a). ChimeraX was used to generate the final renderings shown in Figs. 1b (right) and 4a (right).

## V-ATPase copy number analysis

The V-ATPase maps from wild-type TOMO data 1 were filtered to 30 Å and used as templates for V-ATPase assemblies identification in the wild-type TOMO data 2 and *Syp*[−/−] TOMO data 1 tomograms. Template matching was performed on the binned tomograms (binning factor = 8, 8.8 Å per pixel) with PyTom[65]. Tomograms were then Wiener filtered

using PyCresta (examples are in Extended Data Fig. 8a,b). The Wiener filtering was performed with a defocus setting of 3.0 μm and a signal-to-noise ratio falloff of 0.9. In addition, the tomograms were denoised by cryoCARE[38] without Wiener filtering (an example is in Fig. 4a). The template-matching results were manually checked in both the Wiener-filtered and cryoCARE denoised maps.

Two individuals independently examined two half sets of the tomograms and counted the copy numbers of intact and V0-only V-ATPase assemblies per ISV (Supplementary Tables 4 and 5). For the intact V-ATPase, we examined 1,326 wild-type ISVs and 1,453 $Syp^{-/-}$ ISVs. For the V0-only V-ATPase, we examined a smaller number of ISVs as it is much more time-consuming to identify the smaller V0-only assemblies (106 wild-type ISVs and 188 $Syp^{-/-}$ ISVs, respectively).

To conduct statistical significance tests, we used a bootstrapping statistical procedure by a random resampling with replacement of the observed copy numbers for each ISV for the four groups (Extended Data Fig. 8d,e). This process was repeated 10,000 times for each group, leading to the generation of 10,000 simulated distributions of V-ATPase copy numbers for each group. The mean and standard deviations were calculated for each copy number and group. Statistical significance was assessed using the Student's $t$-test. All the analyses were performed using the R software.

To fit Poisson distributions to the copy numbers for each of the four groups, ISVs with at least one intact or V0-only V-ATPase assemblies were used (Extended Data Fig. 8f–i). Scale factors between the observed copy number and the Poisson distributions, as well as the λ parameters of the Poisson distributions, were estimated using a least squares method implemented in a Python script. For the case of intact V-ATPase assemblies, the Poisson fits predict the number of ISVs without intact V-ATPases and suggest that our observations overestimated the number of ISVs without intact V-ATPases. This degree of overestimation is largely due to the missing wedge effect. Conversely, the missing wedge effect also indicates that we underestimated the number of intact V-ATPases for copy numbers ≥ 1. However, the λ parameter of the fitted Poisson distribution should be independent of the missed fraction of V-ATPases, so it represents the true average copy number of intact V-ATPases per ISV. For the case of V0 assemblies, similar arguments apply. However, there is no overestimation of the number of ISVs without V0 assembly in wild-type ISVs, possibly due to slight uncertainties in counting V0 assemblies. All calculations were performed using the SciPy module from Python (v3.8).

## Single-particle cryo-EM

For SPA data collection, cryo-EM grids with wild-type ISVs and $Syp^{-/-}$ ISVs were imaged using a Titan Krios electron microscope (Thermo Fisher Scientific) equipped with a K3 camera (Gatan) at the Stanford cEMc using the SerialEM automation software[57]. The nominal magnification was ×81,000, resulting in a physical pixel size of 1.1 Å. At each stage position, a group of four or nine holes was imaged using the multiple record setup, and each hole contained three or four imaging spots. A 50-frame movie stack was collected at each imaging spot with a total exposure time of 4.0 s. The dose rate was approximately 15.5 $e^{-}$ $px^{-1}$ $s^{-1}$ with a 0.08 s exposure time per frame; the total dose of one movie stack was 50 $e^{-}$ $Å^{-2}$.

All images of wild-type ISVs were preprocessed in RELION (v3.1)[70]. In total, 21,577 movies were collected and aligned with Motioncor2[59], and CTF parameters were estimated from the average of aligned frames with CTFFIND4[71]. We used Topaz[72] for particle picking. Of particles, 524 of apparent intact V-ATPases were manually picked from 108 micrographs for the training set. Using this training set, Topaz picked 33,094 particles from 4,404 micrographs. We then performed several rounds of 2D and 3D classification with RELION, which returned 4,461 intact V-ATPase candidates. The initial template for 3D classification was obtained from previous work[41]. Topaz was retrained using these particles and returned 321,087 particles from the entire dataset. After several rounds of 2D and 3D classification, 77,779 particles were kept for further analysis. To classify these particles, we adopted a soft mask of the cytoplasmic region of the intact V-ATPase assembly, which resulted in three classes, representing the three states of the V-ATPase. The map quality was further improved by several rounds of 3D refinement, CTF refinement, postprocessing and Bayesian polishing in RELION. Next, these particles were exported into cryoSPARC (v3.2)[73], followed by non-uniform refinement, which resulted in overall resolutions of 4.3 Å for all three states. For the V0-only V-ATPase assemblies, a rescaled V0-only-focused 2D class from the intact V-ATPase 2D classes was used as a template for particle picking in 13,036 micrographs using Topaz. A total of 4,218 particles were selected by several rounds of 2D classification to retrain the Topaz picker. Ultimately, 699,268 particles were picked, followed by a series of sequential steps in RELION: 2D classification, 3D classification, 3D refinement, CTF refinement, postprocessing and Bayesian polishing. The refined particles were subsequently exported into cryoSPARC. After non-uniform refinement (the initial lowpass resolution was set to 12 Å), the overall resolution was 3.8 Å.

All images of $Syp^{-/-}$ ISVs were preprocessed in cryoSPARC (v4.4)[73]. In total, 20,027 movies were collected and aligned with patch motion correction, and CTF parameters were estimated from the average of aligned frames with patch CTF estimation. Again, we used Topaz[72] for V-ATPase particle picking. Of particles, 179 of apparent intact V-ATPase assemblies were manually picked from 327 micrographs for the training set. We used this training set to pick 47,880 particles from 3,500 micrographs, followed by several rounds of 2D classification in cryoSPARC, which returned a total of 10,860 particles. Topaz was retrained using these particles, and it picked 423,510 particles from the entire dataset. After several rounds of 2D classification, 137,987 particles were exported into RELION (v4.0)[74] for further 3D classification. To classify these particles with no bias, we first used a soft mask that was derived from the map of the intact V-ATPase assembly in wild-type ISVs. The resulting 84,686 particles were further classified using a soft mask of the cytoplasmic region of the V-ATPase, resulting in three classes representing three states of the V-ATPase. The map quality was further improved by several rounds of 3D refinement, postprocessing and CTF refinement in RELION, which resulted in three maps with resolutions of 4.5, 4.4 and 4.5 Å for states 1, 2 and 3, respectively. For the V0-only V-ATPase assemblies in $Syp^{-/-}$ ISVs, we used the V0-only Topaz model of the wild-type ISVs dataset and performed particle picking from 13,036 micrographs. A total of 639,377 particles were picked, followed by two rounds of 2D classification. To further classify these particles, we used the 2D classification-accepted 91,542 particles for ab initio reconstruction of four classes, and one decoy class consisting of 2D classification-declined particles. Then, we applied these five classes as hetero-refinement templates of 91,542 particles, which returned 56,885 particles for 3D classification and selected 53,390 particles for further processing in cryoSPARC: non-uniform refinement (the initial lowpass resolution was set to 30 Å for the hetero-refined map), global CTF refinement, local CTF refinement, local refinement and non-uniform refinement (the initial lowpass resolution was set to 12 Å for the local refined map). The final refined map of V0-only V-ATPase assemblies in $Syp^{-/-}$ ISVs had a resolution of 3.6 Å.

All Fourier shell correlation curves were calculated with independently refined half-maps, and the resolution was assessed using the 0.143 criterion with a correction for the masking effects. The local resolutions of all the wild-type maps and V0-only V-ATPase maps of $Syp^{-/-}$ ISVs were estimated by the local resolution estimation method in cryoSPARC. The local resolutions of the intact V-ATPase maps for $Syp^{-/-}$ ISVs were estimated using the local resolution method in RELION. The visualization of the local resolution maps (Extended Data Figs. 2f and 4f) was performed with ChimeraX. Orientational sampling was assessed for the final maps of the intact and V0-only V-ATPase assemblies (Extended Data Figs. 2g and 4g) using the Euler angles from RELION 3D refinement and cryoSPARC non-uniform refinement results.

## Measurement of ISV diameters

Diameters of ISVs were estimated from micrographs collected for SPA and from tomograms. For each tomogram, a 2D projection was generated by summing intensities along the $z$ axis. For each ISV containing at least one detected intact or V0-only V-ATPase, three points on its membrane edge were manually selected. Then, a circle was fit based on these points to calculate the diameter of the ISV.

## Model building and refinement

Initial models of the V-ATPase and synaptophysin complexes for our wild-type (mouse) ISV SPA data were generated based on the deposited rat (PDB IDs: 6VQ6, 6VQ7, 6VQ8 and 6VQH) and human (PDB IDs: 6WM2, 6WM3, 6WM4 and 6WLW) V-ATPase structures and a mouse synaptophysin model predicted by AlphaFold2[29]. After mutation of the model according to the mouse sequence with Coot[75], the models were fitted as rigid bodies into cryo-EM maps using ChimeraX[76]. These models were manually adjusted in Coot before being imported into ISOLDE[77] within ChimeraX to adjust sidechain rotamers. The final models were evaluated through multiple rounds of refinement using Coot and Phenix (v.1.21)[78] and validated with EMRinger[79]. The atomic models for the remaining regions of the V-ATPase assemblies were modelled based on previous work[18], followed by mutation according to the mouse sequence using Coot. The resulting models were then rigid-body fitted into the focused maps by ChimeraX, and then refined with Coot and Phenix. In the final round of real-space refinement with Phenix, we used non-default settings for const_shrink_donor_acceptor=0, nonbonded_weight=500, and use_neutron_distances=true. For illustration purposes, composite models of three rotational states were generated and followed the same strategy as in previous work[18].

The structures of the intact and V0-only V-ATPase assemblies in $Syp^{-/-}$ ISVs were modelled similarly to our wild-type structures, which involved automatic remodelling by Rosetta[80,81], iterative manual adjustment by Coot and real space refinement and ADP refinement with Phenix. In the final round of real-space refinement with Phenix, we used non-default settings for const_shrink_donor_acceptor=0, nonbonded_weight=500, and use_neutron_distances=true.

## Structural analysis

We used the ChimeraX electrostatic function to calculate the electrostatic potential of the interface region of the V-ATPase–synaptophysin complex.

To gain insights into structural variations, we computed the root-mean-square differences (RMSDs) for Cα atoms between all the wild-type and $Syp^{-/-}$ models of both intact and V0-only V-ATPases (Supplementary Table 3). We used state 2 of the intact V-ATPase for PDB IDs 6WM3, 7U4T, 6VQG and 7UNF, as the 7UNF deposition only includes state 2. These models, along with the $Syp^{-/-}$ models, were globally aligned to the wild-type models, and the RMSD values were subsequently calculated using a custom Python script. The RMSDs were visualized in PyMol with the ColorbyRMSD function.

## Proteomics of wild-type ISV

To obtain a comprehensive list of proteins from the ISV sample, we used the high-resolution mass spectrometry sample preparation and liquid chromatography–tandem mass spectrometry (LC–MS/MS) method. In brief, we added 100 µl of 6 M GdmCl, 10 mM TCEP, 40 mM CAA and 100 mM Tris (pH 8.5) buffer to three different ISVs preparations (26 µg each). Lysates were incubated at 95 °C for 5 min and briefly vortexed. The protein concentration was measured using the bicinchoninic acid assay method. Samples were then digested by trypsin overnight at 37 °C with a protein-to-enzyme ratio of 50:1. Digestion was stopped by adding 1% trifluoroacetic acid (TFA) and samples were cleaned up using an Oasis HLB cartridge (1 cc per 10 mg, Waters). Digested peptide samples were dried by speed vac and redissolved in 100 mM triethylammonium bicarbonate (TEAB), and were labelled with tandem mass tag (TMT) 10plex reagent (Thermo Fisher Scientific) as instructed by the vendor and subsequently combined at equal amounts.

Waters 2D LC (Waters MClass 2DnLC) was used for peptide separation. Peptides were separated by reverse-phase chromatography at high pH in the first dimension, followed by an orthogonal separation at low pH in the second dimension. In the first dimension, the mobile phases were buffer A (20 mM ammonium formate at pH 10) and buffer B (acetonitrile). Peptides were separated on a BEH 300 µM × 5 cm C18 5.0-µM column (Waters) using 12 discontinuous step gradients at 2 µl min⁻¹. In the second dimension, peptides were loaded to an in-house packed 75 µM ID/10 µM tip ID × 28 cm C18-AQ 1.8-µM resin column with buffer A (0.1% formic acid in water). Peptides were separated with a gradient from 5% to 40% buffer B (0.1% formic acid in acetonitrile) at a flow rate of 300 nl min⁻¹ in 180 min. The liquid chromatography system was directly coupled in-line with an Orbitrap Fusion Lumos Mass Spectrometer (Thermo Fisher Scientific).

The source was operated at 1.8–2.2 kV to optimize the nanospray with the ion transfer tube at 275 °C. The mass spectrometer was run in a data-dependent mode. A full mass spectrometry scan was acquired in the Orbitrap mass analyzer from 400 to 1,500 $m/z$ with a resolution of 120,000. Precursors were isolated with an isolation window of 0.7 $m/z$ and fragmented using collision-induced dissociation at 35% energy in the ion trap in rapid mode. MS1 automatic gain control (AGC) was $4 \times 10^4$; MS2 AGC was $10^4$. The maximum injection time was 100 ms. Subsequently, eight fragment ions were selected for MS3 analysis, isolated with an $m/z$ window of 1.6, and fragmented with higher-energy collisional dissociation (HCD) at 65% energy. Resulting fragments were detected in the Orbitrap at 60,000 resolution, with a maximum injection time of 150 ms or until the MS3 AGC target value of $10^5$ was reached.

The raw data acquired were processed with the Proteome Discoverer (Thermo Fisher Scientific). A mass tolerance of 10 ppm was used for precursor ions and 0.6 Da for fragment ions for the UniProt *Mus musculus* proteins database search. The search included cysteine carbamidomethylation as a fixed modification. Acetylation at the protein N terminus, methionine oxidation and TMT at the peptide N terminus and lysine were used as variable modifications. Up to two missed cleavages were allowed for trypsin digestion. Only unique peptides with a minimum of six amino acids in length were considered for protein identification. The peptide false discovery rate was set at less than 1%. Data were searched against the mouse database from UniProt. Spectra with more than 50% interference were excluded for subsequent quantitative analysis. The final protein lists (2,687 protein candidates in total; Supplementary Table 1) used two filters: (1) a protein with more than two unique peptides; and (2) it is marked as a master protein.

## Identification of the e subunit

To identify the isoform of V-ATPase subunit e in ISVs, we used chymotrypsin in the mass spectrometry sample preparation. Of each of three different ISV preparations, 5 µg was transferred to a new tube and normalized to 50 µl with 10% SDS/100 mM TEAB. The samples were then reduced with 10 mM dithiothreitol (DTT) for 20 min at 55 °C, cooled to room temperature and then alkylated with 30 mM acrylamide for 30 min. They were then acidified to a pH ~ 1 with 2.6 µl of 27% phosphoric acid and in 165 µl of S-trap loading buffer (90% methanol/10% 1 M TEAB) and loaded onto S-trap microcolumns (Protifi). After loading, the samples were washed sequentially with 150-µl increments of 90% methanol/10% 100 mM TEAB, 90% methanol/10% 20 mM TEAB and 90% methanol/10% 5 mM TEAB solutions, respectively. Samples were digested at 47 °C for 2 h with 600 ng of mass spectrometry-grade chymotrypsin (Promega). The digested peptides were then eluted with two 35-µl increments of 0.2% formic acid in water and two more 40-µl increments of 80% acetonitrile with 0.2% formic acid in water.

The four eluents were consolidated in 1.5-ml S-trap recovery tubes and dried via SpeedVac (Thermo Fisher Scientific). Finally, the dried peptides were reconstituted in 2% acetonitrile with 0.1% formic acid in water for LC–MS analysis.

Proteolytically digested peptides were separated using an in-house pulled and packed reversed-phase analytical column (approximately 25 cm long, 100 μm of inner diameter), with Dr. Maisch 1.9-μm C18 beads as the stationary phase. Separation was performed with an 80-min reverse-phase gradient (2–45% B, followed by a high-B wash) on an Acquity M-Class UPLC system (Waters Corporation) at a flow rate of 300 nl min$^{-1}$. Mobile phase A was 0.2% formic acid in water, whereas mobile phase B was 0.2% formic acid in acetonitrile. Ions were formed by electrospray ionization and analysed by an Orbitrap Exploris 480 mass spectrometer (Thermo Fisher Scientific). The mass spectrometer was operated in a data-dependent mode using HCD fragmentation for MS/MS spectra generation.

The raw data were analysed using Byonic v5.1.1 (Protein Metrics) to identify peptides and infer proteins. Concatenated FASTA files containing UniProt *Mus musculus* proteins and other likely contaminants and impurities were used to generate an in silico peptide library. Proteolysis with chymotrypsin was assumed to be semi-specific, allowing for N-ragged cleavage with up to two missed cleavage sites. The precursor and fragment ion tolerances were set to 12 ppm. Cysteine modified with propionamide was set as a fixed modification in the search. Variable modifications included oxidation on methionine, deoxidation on tryptophan, glutamine and glutamic acid cyclization and N-terminal acetylation. Proteins were held to a false discovery rate of 1% using the standard reverse-decoy technique[82]. The final identified protein lists (Supplementary Table 2) used two filters: (1) the identified peptide is unique; and (2) the spectrum score is larger than 300 according to Byonic's scoring algorithm[83].

## Unbiased matching of the binding partner density

The binding partner densities were manually extracted from both the SPA maps of the intact (state 3) or the V0-only V-ATPase assembly of wild-type ISVs using UCSF Chimera. The extracted maps were placed in a 100 px$^3$ box with a pixel size of 1.1 Å. AlphaFold2[29] predicted atomic models corresponding to the high-resolution mass spectrometry-detected ISV proteins were downloaded. The unbiased matching was performed using the CoLoRes program in the Situs package[30,31] with a resolution setting of 6 Å, a searching degree of 15, and a map contour cut-off of 0.01 or 0.004 for the intact or V0-only segmented extra density map, respectively. The matching was scored and ranked by unnormalized CoLoRes cross-correlation scores (CCSs) (rows 2–2,641 of Supplementary Table 1). There are 49 other proteins (rows 2,642–2,694 of Supplementary Table 1) without an AlphaFold2 prediction, but they are either too large in size or not known to be synaptic membrane proteins. Moreover, some large structures that led to unrealistically high CCS values (more than 1) were checked and ignored. The top 200 hits with reasonable CCS values (less than 1) were all inspected individually with the target densities in ChimeraX. Candidates that were not membrane proteins or had no membrane domain matching the observed binding partner density in the membrane region were ignored for further analysis. Among those that appeared reasonable, we rigid-body adjusted the fit using ChimeraX. We then calculated the percentage of outliers using ChimeraX by normalizing the number of Cα atoms residing outside the binding partner density against the total number of Cα atoms. Using the percentage of outliers as a criterion, the final set of top candidates was SYG1, synatogyrin-3, synatoporin and synaptophysin.

## Pharmacological seizure susceptibility

Mice of each genotype were assayed between 4 and 6 months of age. All mice were weighed on the day of the experiment and were administered 25 mg kg$^{-1}$ kainic acid intraperitoneally from a freshly prepared 5 mg ml$^{-1}$ PBS solution (*n* = 9 wild-type and *n* = 6 *Syp*$^{-/-}$ mice). Mice

were immediately placed in a cylindrical observation chamber and monitored and scored in real time. The observational behaviour was scored blind according to the modified Racine scale 6 or 8 as previously defined[84]. The latency curves were compared using a survival log-rank (Mantel–Cox) test in GraphPad Prism.

## Software and code

We used AlphaFold2[29], Amira (v2020.2; Thermo Fisher Scientific), AreTomo (v1.3.4)[61], Byonic by ProteinMetrics (v5.1.1)[83], Chimera (v1.16) and ChimeraX (v1.3 and v1.7)[76], cryoSPARC (v3.2 and v4.4)[73], CTFFIND4[71], iBright1500 analysis software (v5.2.2; Thermo Fisher Scientific), EMRinger (v1.0.0)[79], IMOD (v4.11.3)[60], ISOLDE (v1.3)[77], Motioncor2[59], Phenix (v1.21)[78], Prism (v10.2.2; GraphPad Software), Proteome Discoverer (v2.1; Thermo Fisher Scientific), PyCrESTA (https://github.com/brungerlab/pycresta), PyMol (v2.3.2; Schrödinger, LLC), Python (v3.8), PySeg (v1.0.0)[63], PyTom (v0.981a and v1.1)[65], R (v3.4.2), RELION (v2, v3.1 and v4.0)[70], Rosetta[80], Scientific Xcalibur (v4.1; Thermo Fisher Scientific), SerialEM (v4.0 and v4.1)[57], Situs (v3.2)[31], Topaz (v0.2.5)[72] and Warp (v1.0.9)[66]. Several script files are available at https://github.com/brungerlab/ISV_scripts.

## Animal statement

All animal procedures were performed in accordance with the National Institutes of Health Guide for the Care and Use of Laboratory Animals and approved by the Stanford Administrative Panel on Laboratory Animal Care institutional guidelines (protocol 29981) and by the University of Colorado Boulder Institutional Animal Care and Use Committee (protocol 1106.02).

## Statistics and reproducibility

No statistical method was used to predetermine the sample size, but experiments described in this study were performed with at least 3–9 samples for each group. The bootstrapping statistical procedures were done by randomly sampling repeats with replacement (Extended Data Fig. 8d,e). Two individuals independently examined two halves of the tomograms and counted the copy numbers of V-ATPase per ISV (Fig. 4b,c). The mice observational behaviour was scored blind by two individuals (Fig. 4d).

## Reporting summary

Further information on research design is available in the Nature Portfolio Reporting Summary linked to this article.

## Data availability

The subtomogram averaging maps (44858 (WT V0 only), 44855 (WT state 1), 44856 (WT state 2) and 44857 (WT state 3)), the SPA maps (44846 (WT V0 only), 44843 (WT state 1), 44839 (WT state 2), 44840 (WT state 3), 44845 (*Syp*$^{-/-}$ V0 only), 44844 (*Syp*$^{-/-}$ state 1), 44842 (*Syp*$^{-/-}$ state 2) and 44841 (*Syp*$^{-/-}$ state 3)), and representative binned tomograms (44847 (*Syp*$^{-/-}$ ISV) and 44848 (WT ISV)) have been deposited in the Electron Microscopy Data Bank. The atomic coordinates have been deposited in the Protein Data Bank (9BRZ (WT V0 only), 9BRT (WT state 1), 9BRA (WT state 2), 9BRQ (WT state 3), 9BRY (*Syp*$^{-/-}$ V0 only), 9BRU (*Syp*$^{-/-}$ state 1), 9BRS (*Syp*$^{-/-}$ state 2) and 9BRR (*Syp*$^{-/-}$ state 3)). Source data are provided with this paper.

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

**Acknowledgements** We thank T. Südhof, D. Baker and P. Sturmfels for stimulating discussions; M. Synder for discussions and access to mass spectrometry instrumentation; R. Pfuetzner and N. Gopal for peptide and mouse preparations; R. Leube for providing the *Syp*⁻/⁻ mice; J. Matsuda for mouse line recovery; X. Chen and K. Grounds for assistance with mouse breeding; P. Villar and M. Amini for assistance with seizure experiments; S. Klumpe for help with model building; E. Montabana and C. Zhang for help with the electron microscopy data collection at the Stanford University cEMc; J. Shen for helpful comments; and Paul D. Adams, Pavel Afonine, Nigel Moriarty, K. Ian White for help with Phenix refinement. Funding was provided by the US National Institutes of Health (NIH; RO1MH63105 to A.T.B.; NIH EUREKA 1R01GM114496-01 to M.H.B.S.; and T32 GM065103)/CU Molecular Biophysics Predoctoral Traineeship to D.J.A.), the Beijing Natural Science Foundation (JQ24031 to Q.G.), the MCDB Neurodegenerative Disease Fund (to M.H.B.S.), a Howard Hughes Medical Institute CIA award (to M.H.B.S.), a Corden Pharma Fellowship (to D.J.A.), the Stanford ADRC Zaffaroni Alzheimer's Disease Translational Research Program (to A.T.B.), and the National Natural Science Foundation of China (#32371191 to Q.G.). The subunit identification by mass spectrometry is supported by the Vincent Coates Foundation Mass Spectrometry Laboratory, Stanford University Mass Spectrometry (RRID:SCR_017801) utilizing the Thermo Exploris 480 nanoLC/MS system (RRID:SCR_022215), and in part by NIH P30 CA124435 utilizing the Stanford Cancer Institute Proteomics/Mass Spectrometry Shared Resource, in particular, G. McKenzie. The contents of this publication are solely the responsibility of the authors and do not necessarily represent the official views of the NIGMS or NIH. This article is subject to HHMI's Open Access to Publications policy. HHMI laboratory heads have previously granted a non-exclusive CC BY 4.0 license to the public and a sublicensable license to HHMI in their research articles. Pursuant to those licenses, the author-accepted manuscript of this article can be made freely available under a CC BY 4.0 license immediately upon publication.

**Author contributions** C.W., W.J., J.L., M.H.B.S., W.B., Q.G. and A.T.B. designed the experiments. C.W., W.J., J.L., L.E., X.W., D.J.A., T.B., L.H. and Q.G. performed the experiments and analysis. K.Y. and R.G.H. assisted with electron microscopy data collection and analysis. X.S., L.J. and R.J. assisted with high-resolution mass spectrometry proteomics data collection and analysis. C.W., W.J., Q.G. and A.T.B. wrote the manuscript.

**Competing interests** The authors declare no competing interests.

**Additional information**
**Correspondence and requests for materials** should be addressed to Qiang Guo or Axel T. Brunger.

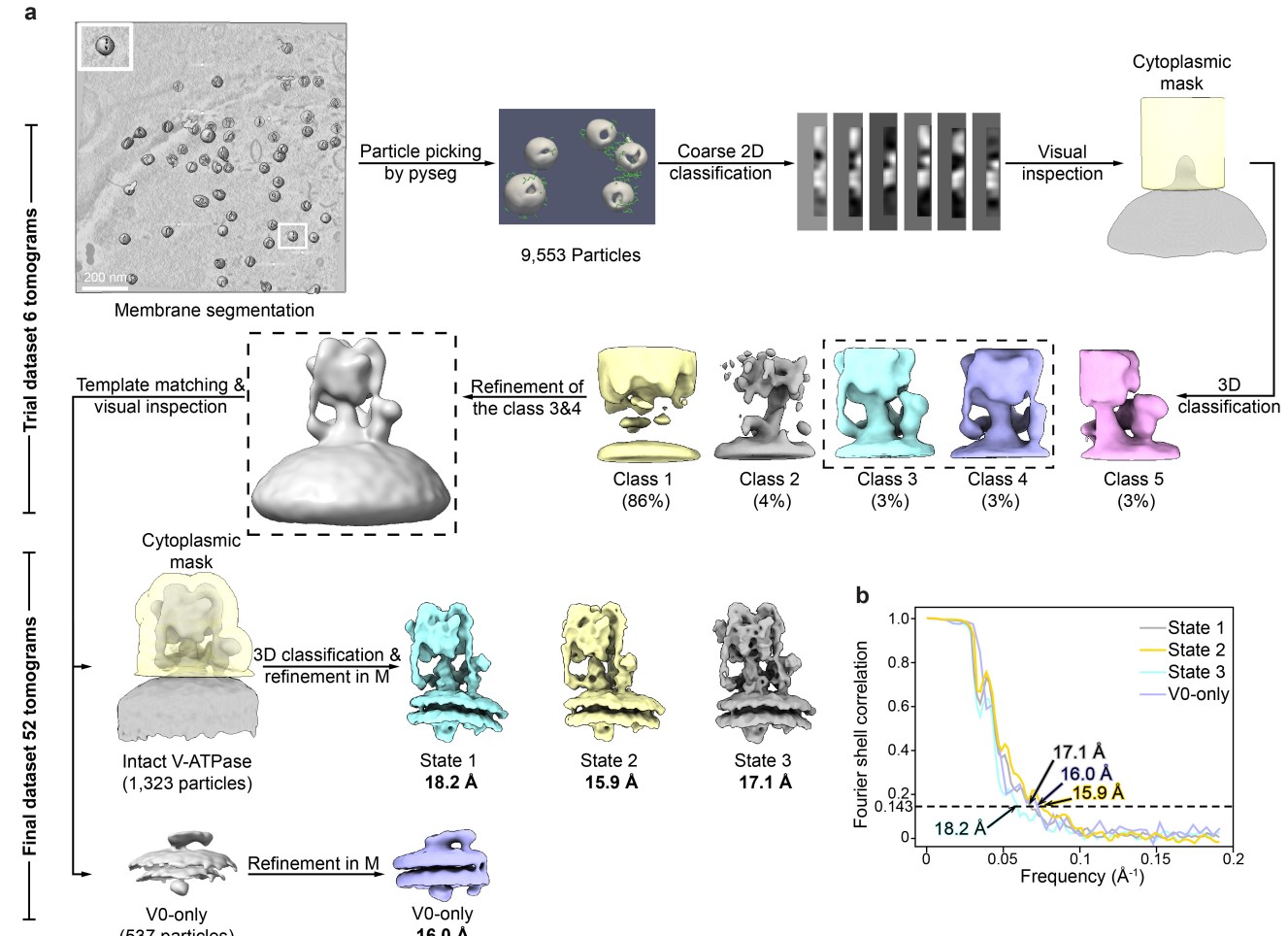

**Extended Data Fig. 1 | Workflow of subtomogram averaging of V-ATPase assemblies in wildtype ISVs (related to Fig. 1). a**, Workflow of particle picking, subtomogram averaging, and classification of intact and V0-only V-ATPase assemblies (see Methods). **b**, Gold-standard Fourier shell correlation (FSC) curves of the resulting subtomogram average maps.

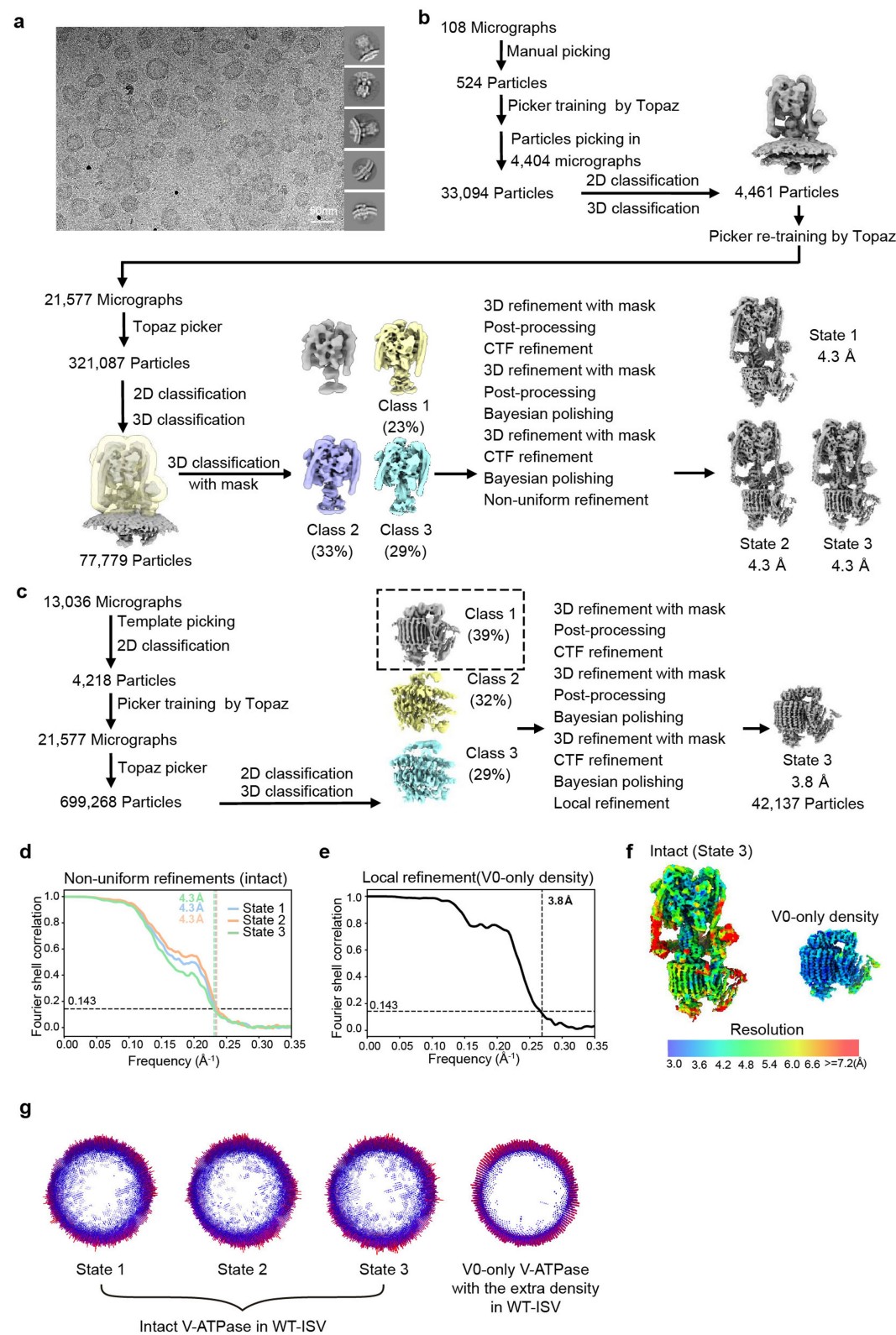

**Extended Data Fig. 2 | Single-particle averaging workflow of V-ATPase assemblies in wild-type ISVs (related to Fig. 3). a**, Representative micrograph (one of 21,577 micrographs) of ISVs and V-ATPase assemblies 2D classification averages. The micrograph was Topaz-denoised for better visualization. **b&c**, Cryo-EM image processing workflow of (**b**) intact V-ATPase assemblies in three rotational states and (**c**) V0-only V-ATPase assemblies (see Methods). **d&e**, Gold-standard FSC curves of the intact V-ATPase assembly in three rotational states and of the V0-only V-ATPase assembly. **f**, Refined maps of intact V-ATPase State 3 and V0-only V-ATPase assemblies, colored according to local resolution estimated in cryoSPARC (see Methods). **g**, Particle orientation distributions of intact V-ATPase assemblies in three rotational states and V0-only V-ATPase assemblies (see Methods). The orientational distributions should adequately sample Fourier space.

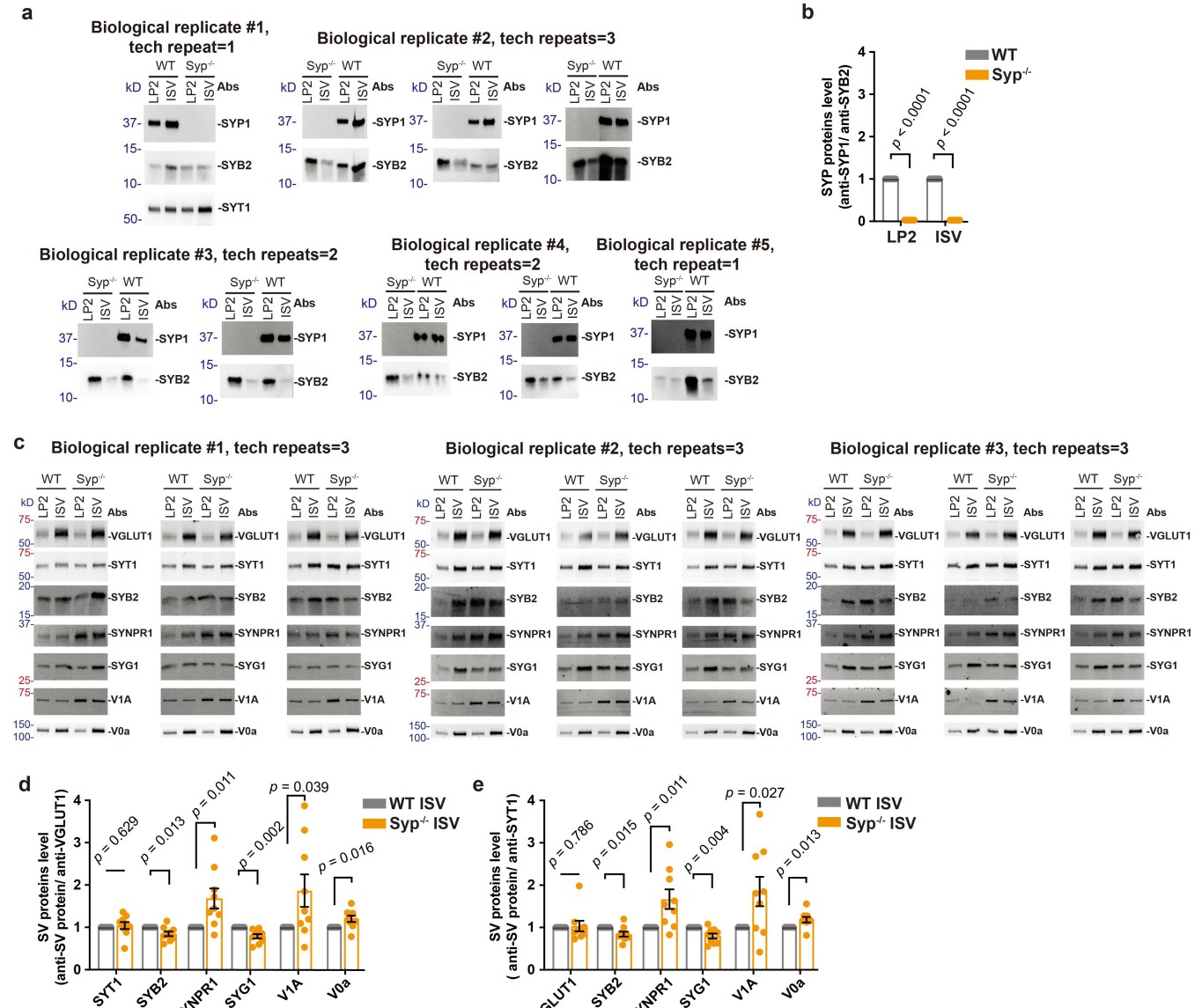

**Extended Data Fig. 3 | Western blots of synaptic proteins in wild-type and** *Syp*$^{-/-}$ **samples (related to Fig. 2). a&b,** Western blots of (**a**) synaptophysin in LP2 and ISV samples from wild-type and *Syp*$^{-/-}$ mouse brains, and (**b**) quantification. Blotting was performed with antibodies (Abs) for synaptophysin-1 (SYP1) (for *Syp*$^{-/-}$ validation), synaptobrevin-2 (SYB2), and synaptotagmin-1 (SYT1). (**b**) For each antibody and blot repeat, the band densities for *Syp*$^{-/-}$ LP2 and *Syp*$^{-/-}$ ISVs were first normalized to the respective band densities of wild-type LP2 and wild-type ISV, respectively. The synaptophysin-1 density of *Syp*$^{-/-}$ LP2 and *Syp*$^{-/-}$ ISV was then normalized to the synaptobrevin-2 density of wild-type LP2 and wild-type ISV, respectively. Data are presented as mean ± SEM; error bars (black) represent the SEM of nine independent measurements (biological and technical repeat numbers are specified in the figure); *p*-values are indicated in the figure and calculated by unpaired two-tailed t-test (n = 9), *** *p* < 0.001. **c&d,** Western blots of (**c**) synaptic proteins in LP2 and ISV samples from wild-type and *Syp*$^{-/-}$

ISVs mouse brains, followed by quantification (**d**, **e**). Blotting was performed with antibodies (Abs) for synaptotagmin-1 (SYT1), vesicular glutamate transporter-1 (VGLUT1), synaptoporin-1 (SYNPR1), synaptogyrin-1 (SYG1), subunit A of the V-ATPase V1 assembly (V1A), and subunit a of the V-ATPase V0 assembly (V0a). (**e**) For each antibody and blot repeat, the band densities for *Syp*$^{-/-}$ ISV were first normalized to the respective band densities of wild-type ISVs. The synaptic protein densities of *Syp*$^{-/-}$ ISVs were then normalized to the SYT1 (**d**) or VGLUT1 (**e**) densities in wild-type ISVs, respectively. Data are presented as mean ± SEM; error bars (black) represent the SEM of nine independent measurements (biological and technical repeat numbers are specified in the figure); *p*-values are indicated in the figure and calculated by unpaired two-tailed t-test (n = 9), **p* < 0.05, ** *p* < 0.01. Unprocessed blots and source numerical data are available in Source Data.

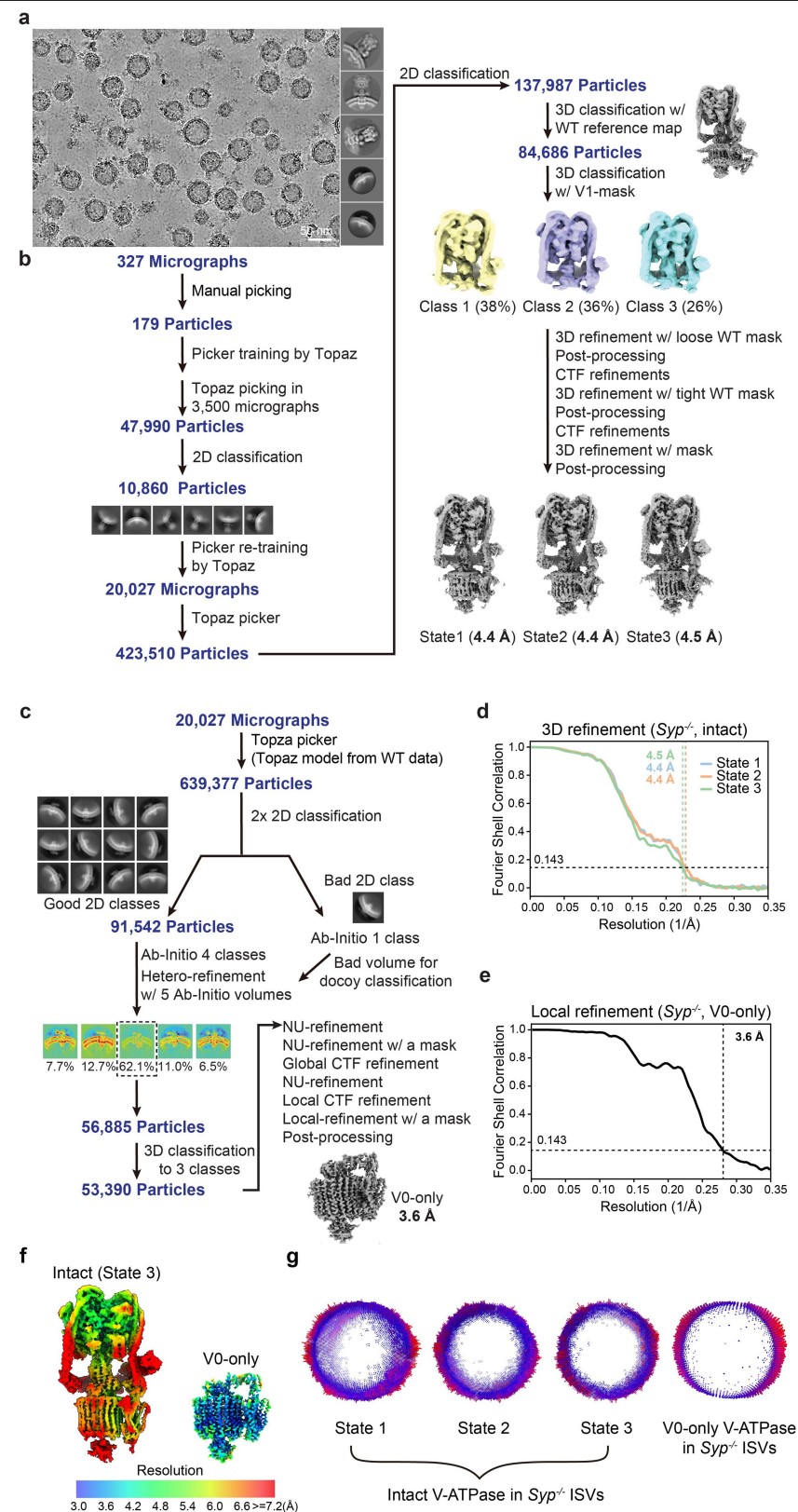

**Extended Data Fig. 4 | Single-particle averaging workflow of V-ATPase assemblies in *Syp*⁻/⁻ ISVs. a**, Representative micrograph (one of 20,027 micrographs) of *Syp*⁻/⁻ ISVs and V-ATPase 2D classification averages. The micrograph was Topaz-denoised for better visualization. **b&c**, Cryo-EM image processing workflow of (**b**) intact V-ATPase assemblies in three rotational states and (**c**) V0-only V-ATPase assemblies in *Syp*⁻/⁻ ISVs (see Methods). **d&e**, Gold-standard FSC curves of intact V-ATPase assemblies in three rotational states and V0-only V-ATPase assemblies in *Syp*⁻/⁻ ISVs. **f**, Refined maps of intact V-ATPase State 3 and V0-only V-ATPase assemblies colored according to local resolution estimated in RELION and cryoSPARC (see Methods). **g**, Orientation distributions for the particles of intact V-ATPase assemblies in three rotational states and V0-only V-ATPase assemblies in *Syp*⁻/⁻ ISVs (see Methods).

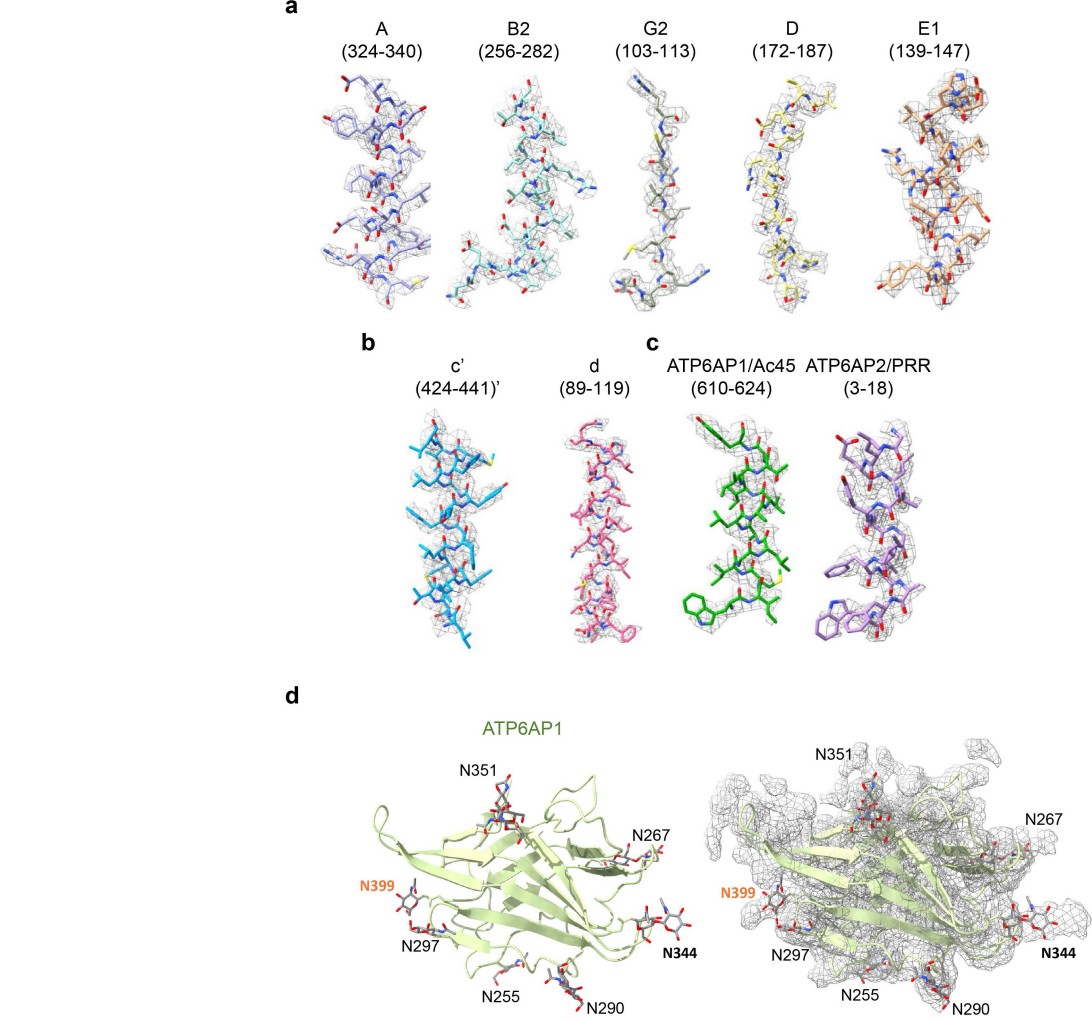

**Extended Data Fig. 5 | Examples of map quality and atomic model fits (related to Fig. 3). a**, Models of subunits A, B2, G2, D, and E1 superimposed on the wild-type intact V-ATPase map. **b**, Models of subunits c′ and d superimposed on the wild-type intact V-ATPase map. **c**, Models of subunits ATP6AP1/AC45 and ATP6AP2/PRR superimposed on the wild-type intact V-ATPase map. **d**, Model of the ATP6AP1 luminal domain superimposed on the wild-type V0-only V-ATPase map. Potential glycosylated sites are shown and annotated using a stick model.

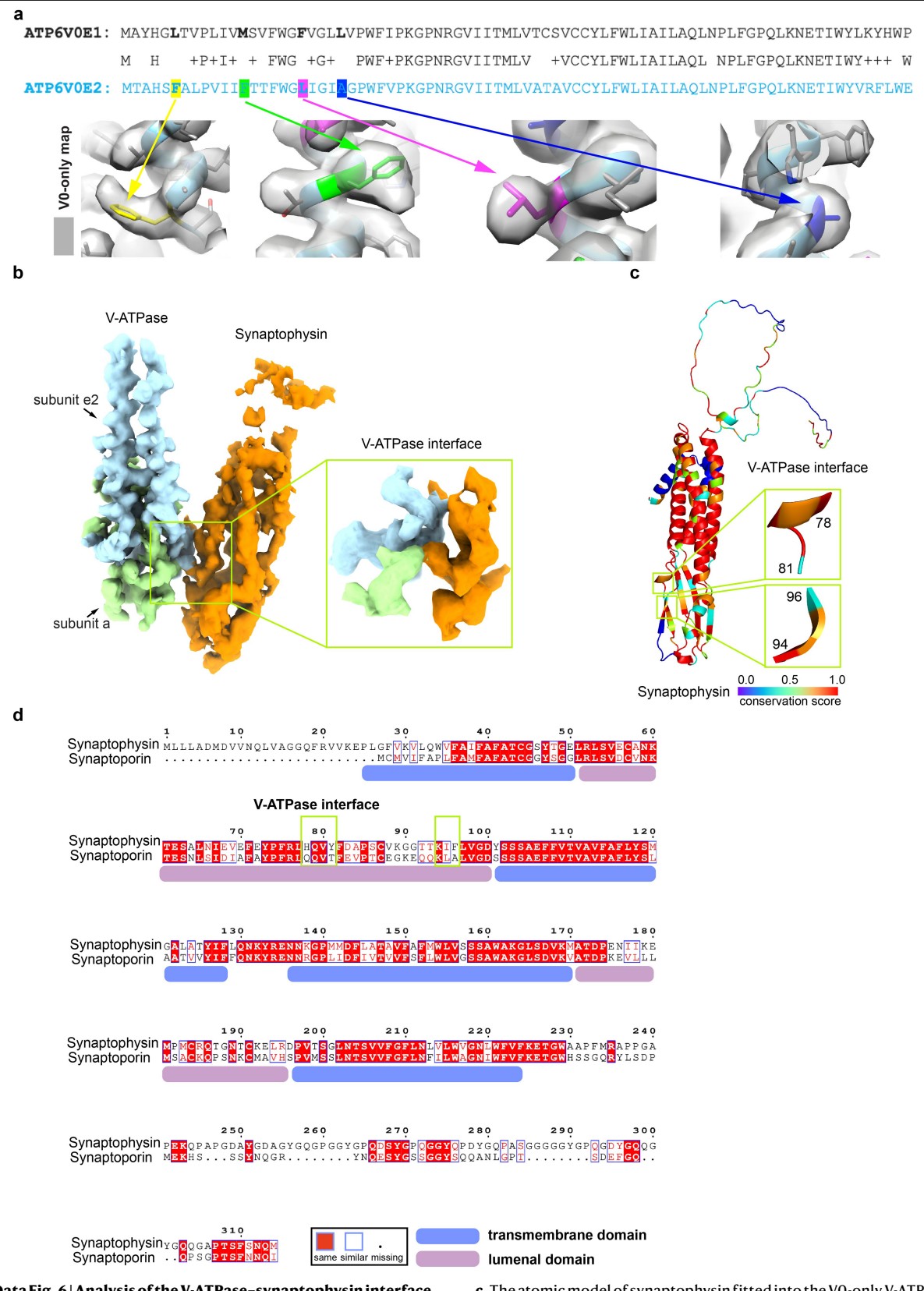

**Extended Data Fig. 6 | Analysis of the V-ATPase–synaptophysin interface (related to Fig. 3). a**, Upper: primary sequence alignment of V-ATPase subunit e1 (ATP6V0E1) and e2 (ATP6V0E2). Representative residues of V-ATPase subunit e2 that differ from e1 are highlighted. Lower: Superposition of the structure and the map for the V0-only V-ATPase assembly in *Syp*$^{-/-}$ ISVs (we used the *Syp*$^{-/-}$ data since it produced the highest resolution for the V0 domain). **b**, SPA map of subunits e2 (blue), a (green) and synaptophysin (orange) for the V0-only V-ATPase assembly of wild-type ISVs. The inset shows a close-up view.

**c**, The atomic model of synaptophysin fitted into the V0-only V-ATPase map of wild-type ISVs (see Methods) is colored according to the primary sequence conservation score compared with synaptoporin. The inset shows a close-up view. **d**, Primary sequence alignment of synaptophysin and synaptoporin. Red-shaded regions indicate strictly conserved residues, and blue boxes indicate conserved residues. Thick blue and pink stripes indicate the transmembrane and luminal domains of synaptophysin and synaptoporin, respectively. The V-ATPase–synaptophysin interface region is annotated with a green box.

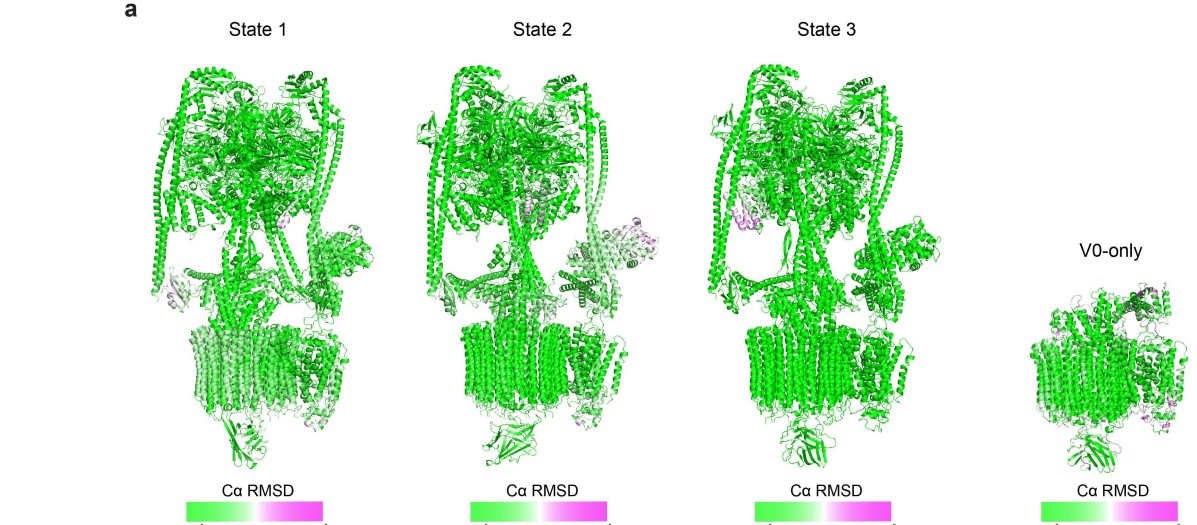

State 1                    State 2                    State 3                    V0-only

Cα RMSD                    Cα RMSD                    Cα RMSD                    Cα RMSD
0 Å          15 Å          0 Å          15 Å          0 Å          15 Å          0 Å          5 Å

**Extended Data Fig. 7 | Comparison of V-ATPase structures in wild-type and *Syp*⁻/⁻ ISVs (related to Fig. 3). a**, Superposition of wild-type and *Syp*⁻/⁻ intact State 1, State 2, State 3, and V0-only V-ATPase assemblies (from left to right). Models were aligned globally. Most of the larger differences are in regions that are poorly determined in the maps.

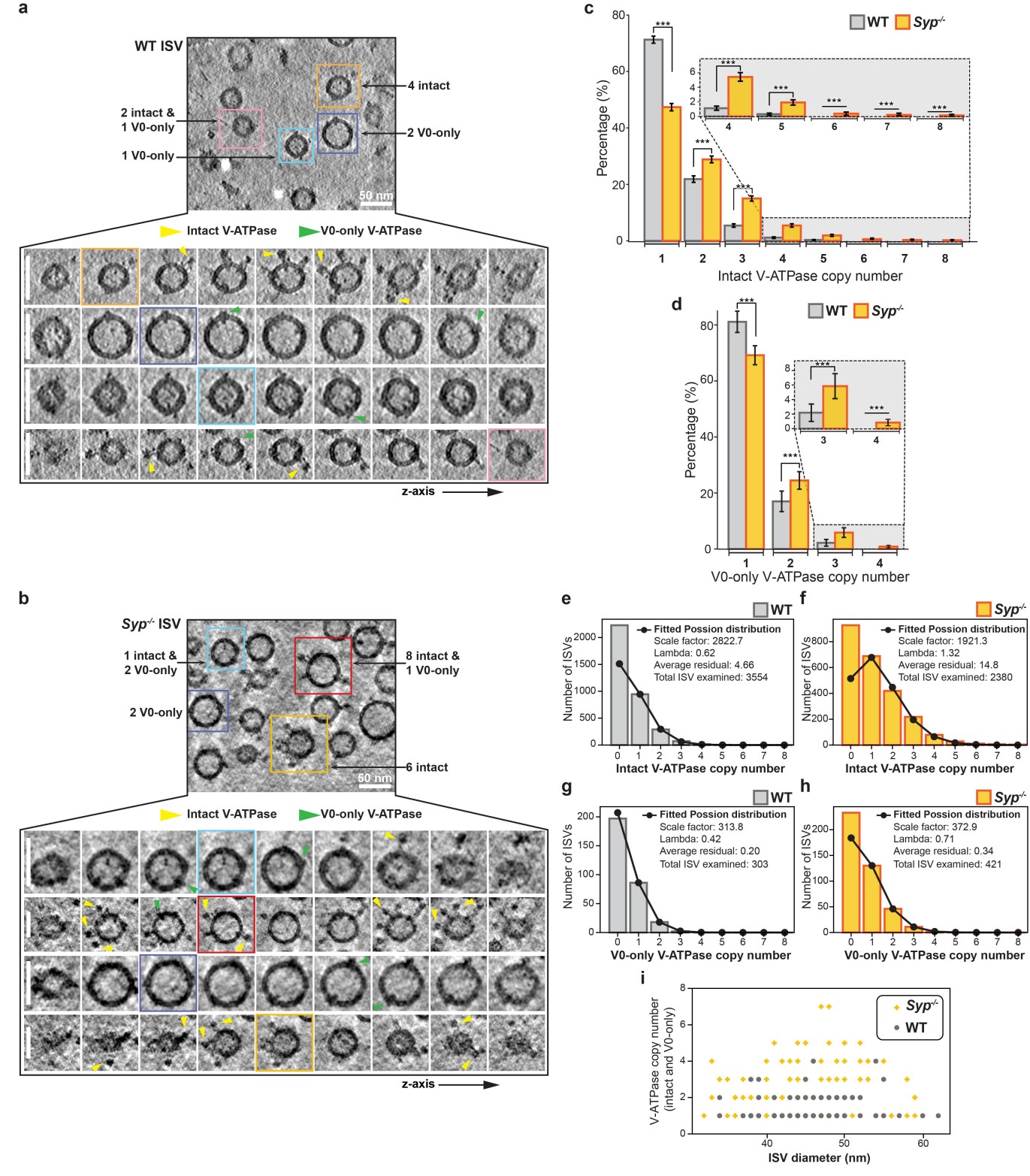

**Extended Data Fig. 8** | See next page for caption.

**Extended Data Fig. 8 | V-ATPase copy numbers from wild-type and *Syp*<sup>-/-</sup> mouse brains (related to Fig. 4). a&b**, Representative tomographic slices of *Syp*<sup>-/-</sup> ISVs (one of 78 tomograms) (**a**, upper) or wild-type ISVs (one of 156 tomograms) (**b**, bottom) (thickness = 0.9 nm), respectively (obtained from Wiener filtered tomograms). Four representative ISVs are selected (colored boxed), and a series of higher magnification tomographic Wiener-filtered planes of each ISV are shown below and annotated with yellow and green arrows for intact and V0-only V-ATPase assemblies, respectively. Scale bars: 50 nm. **c**, Scatter plot of the ISV diameters and V-ATPase copy numbers (including both intact and V0-only assemblies) for wild-type and *Syp*<sup>-/-</sup> ISVs from 6 randomly selected tomograms, respectively. **d&e**, Bootstrapping statistical significance tests (see Methods) for intact (**c**) or V0-only (**d**) V-ATPase copy number per ISV from wild-type and *Syp*<sup>-/-</sup> mouse brains. The data were resampled 10,000 times with replacement to estimate the mean and standard deviation of each intact or V0-only V-ATPase assembly copy number group.

For the intact V-ATPase assembly (Supplementary Table 4), the wild-type ISV and *Syp*<sup>-/-</sup> ISV sample pool comprises 1,326 ISVs and 1,453 ISVs, respectively. For the V0-only V-ATPase assembly (Supplementary Table 5), the wild-type ISV and *Syp*<sup>-/-</sup> ISV sample pool comprises 106 ISVs and 188 ISVs, respectively. Data are presented as mean ± SD; error bars (black) represent the SD of 10,000 statistically independent sampling; *** $p < 0.001$ by Student's t-test (n = 10,000). **f, g, h & i**, Fitted Poisson distribution for the copy numbers of intact V-ATPase assemblies for ISVs from wild-type (**f**) and *Syp*<sup>-/-</sup> mouse brains (**g**), and to the copy numbers of the V0-only V-ATPase assemblies for ISVs from wild-type (**h**) and *Syp*<sup>-/-</sup> mouse brains (**i**). The Poisson distributions were fitted to the observed copy numbers ≥ 1, *i.e.*, the copy numbers of ISVs without any V-ATPase were not used for the fit (see Methods). The least-squares residuals shown in the figure legends refer to copy numbers ≥ 1. Source numerical data are in Source Data.

**Extended Data Table 1 | CryoET data and map statistics**

| Data collection | WT TOMO data1 wild-type ISVs | | | | WT TOMO data2 wild-type ISVs | *Syp*⁻/⁻ TOMO data1 *Syp*⁻/⁻ ISVs |
|---|---|---|---|---|---|---|
| Magnification | 53,000× | | | | 81,000× | |
| EM & Voltage (kV) | Titan Krios, 300 | | | | Titan Krios, 300 | |
| Total dose (e⁻/Å²) | 120 | | | | 139 | |
| Pixel size (Å/pix) | 2.6 | | | | 1.1 | |
| Dose rate (e⁻/px/s) | 6.09 | | | | 5.27 | |
| Defocus range (μm) | 4.0 to 5.0 | | | | 1.0 to 4.0 | |

| Data processing | V-ATPase | | | | | |
|---|---|---|---|---|---|---|
| | V0-only | State 1 | State 2 | State 3 | | |
| Symmetry imposed | C1 | | | | | |
| FSC threshold | 0.143 | | | | | |
| EMDB ID | 44858 | 44855 | 44856 | 44857 | | |
| Particle images contribution to maps | 537 | 317 | 542 | 463 | | |
| Global resolution (FSC = 0.143, Å) | 16 | 18.2 | 15.9 | 17.1 | | |

**Extended Data Table 2 | Cryo-EM data collection, refinement, and validation statistics**

| | V-ATPases in wild-type ISVs | | | | V-ATPases in *Syp*[-/-] ISVs | | | |
|---|---|---|---|---|---|---|---|---|
| | #1 Intact State 1 (EMDB-44843) (PDB 9BRT) | #2 Intact State 2 (EMDB-44839) (PDB 9BRA) | #3 Intact State 3 (EMDB-44840) (PDB 9BRQ) | #4 V0-only (EMDB-44846) (PDB 9BRZ) | #5 Intact State 1 (EMDB-44844) (PDB 9BRU) | #6 Intact State 2 (EMDB-44842) (PDB 9BRS) | #7 Intact State 3 (EMDB-44841) (PDB 9BRR) | #8 V0-only (EMDB-44845) (PDB 9BRY) |
| **Data collection and processing** | | | | | | | | |
| Magnification | 81,000× | 81,000× | 81,000× | 81,000× | 81,000× | 81,000× | 81,000× | 81,000× |
| Voltage (kV) | 300 | 300 | 300 | 300 | 300 | 300 | 300 | 300 |
| Electron exposure (e–/Å$^2$) | 50 | 50 | 50 | 50 | 50 | 50 | 50 | 50 |
| Defocus range (μm) | -1.0 to -4.0 | -1.0 to -4.0 | -1.0 to -4.0 | -1.0 to -4.0 | 0.8 to 2.0 | 0.8 to 2.0 | 0.8 to 2.0 | 0.8 to 2.0 |
| Pixel size (Å) | 1.1 | 1.1 | 1.1 | 1.1 | 1.1 | 1.1 | 1.1 | 1.1 |
| Symmetry imposed | C1 | C1 | C1 | C1 | C1 | C1 | C1 | C1 |
| Initial particle images (no.) | 321,087 | 321,087 | 321,087 | 699,268 | 423,510 | 423,510 | 423,510 | 639,377 |
| Final particle images (no.) | 17,889 | 25,667 | 34,536 | 42,137 | 32,181 | 30,487 | 22,018 | 53,390 |
| Map resolution (Å) | 4.3 | 4.3 | 4.3 | 3.8 | 4.5 | 4.4 | 4.4 | 3.6 |
| FSC threshold | 0.143 | 0.143 | 0.143 | 0.143 | 0.143 | 0.143 | 0.143 | 0.143 |
| | | | | | | | | |
| **Refinement** | | | | | | | | |
| Initial model used (PDB code) | 6VQ6, 6WM2, Synaptophysin from AlphaFold2 | 6VQ7, 6WM3, Synaptophysin from AlphaFold2 | 6VQ8, 6WM4, Synaptophysin from AlphaFold2 | 6VQH, 6WLW, Synaptophysin from AlphaFold2 | Model of wild-type V-ATPase | Model of wild-type V-ATPase | Model of wild-type V-ATPase | Model of wild-type V-ATPase |
| Model resolution (Å) | 4.8 | 4.6 | 6.2 | 4.3 | 7.0 | 6.9 | 6.6 | 4.3 |
| FSC threshold | 0.5 | 0.5 | 0.5 | 0.5 | 0.5 | 0.5 | 0.5 | 0.5 |
| Map sharpening $B$ factor (Å$^2$) | -62.2 | -66.7 | -61.2 | -98.7 | -127 | -113 | -112 | -68 |
| Model composition | | | | | | | | |
| Non-hydrogen atoms | 66,937 | 66,859 | 67,168 | 25,204 | 65,549 | 65,548 | 65,594 | 23,530 |
| Protein residues | 8,563 | 8,554 | 8,590 | 3,277 | 8,388 | 8,388 | 8,393 | 3,066 |
| Ligands | 0 | 0 | 0 | 12 | 0 | 0 | 0 | 12 |
| $B$ factors (Å$^2$) | | | | | | | | |
| Protein | 225.59 | 205.58 | 243.03 | 139.95 | 84.43 | 80.38 | 94.84 | 72.49 |
| R.m.s. deviations | | | | | | | | |
| Bond lengths (Å) | 0.004 | 0.005 | 0.004 | 0.006 | 0.004 | 0.005 | 0.005 | 0.005 |
| Bond angles (°) | 0.734 | 0.811 | 0.830 | 0.924 | 0.773 | 0.814 | 0.835 | 0.876 |
| Validation | | | | | | | | |
| MolProbity score | 1.37 | 1.56 | 1.56 | 2.04 | 1.44 | 1.58 | 1.54 | 1.91 |
| Clashscore | 2.43 | 4.23 | 3.94 | 5.44 | 2.88 | 3.89 | 3.84 | 5.08 |
| Poor rotamers (%) | 0.01 | 0.10 | 0.15 | 3.59 | 0.04 | 0.09 | 0.42 | 2.88 |
| | | | | | | | | |
| Ramachandran plot | | | | | | | | |
| Favored (%) | 95.08 | 94.83 | 94.41 | 95.28 | 94.92 | 94.03 | 94.69 | 95.81 |
| Allowed (%) | 4.90 | 5.15 | 5.49 | 4.69 | 5.04 | 5.88 | 4.95 | 4.16 |
| Disallowed (%) | 0.02 | 0.02 | 0.11 | 0.03 | 0.04 | 0.08 | 0.36 | 0.03 |

# Reporting Summary

## Statistics

For all statistical analyses, confirm that the following items are present in the figure legend, table legend, main text, or Methods section.

| n/a | Confirmed | |
|---|---|---|
| ☐ | ☒ | The exact sample size (*n*) for each experimental group/condition, given as a discrete number and unit of measurement |
| ☐ | ☒ | A statement on whether measurements were taken from distinct samples or whether the same sample was measured repeatedly |
| ☐ | ☒ | The statistical test(s) used AND whether they are one- or two-sided *Only common tests should be described solely by name; describe more complex techniques in the Methods section.* |
| ☒ | ☐ | A description of all covariates tested |
| ☐ | ☒ | A description of any assumptions or corrections, such as tests of normality and adjustment for multiple comparisons |
| ☐ | ☒ | A full description of the statistical parameters including central tendency (e.g. means) or other basic estimates (e.g. regression coefficient) AND variation (e.g. standard deviation) or associated estimates of uncertainty (e.g. confidence intervals) |
| ☐ | ☒ | For null hypothesis testing, the test statistic (e.g. *F*, *t*, *r*) with confidence intervals, effect sizes, degrees of freedom and *P* value noted *Give P values as exact values whenever suitable.* |
| ☒ | ☐ | For Bayesian analysis, information on the choice of priors and Markov chain Monte Carlo settings |
| ☒ | ☐ | For hierarchical and complex designs, identification of the appropriate level for tests and full reporting of outcomes |
| ☒ | ☐ | Estimates of effect sizes (e.g. Cohen's *d*, Pearson's *r*), indicating how they were calculated |

*Our web collection on statistics for biologists contains articles on many of the points above.*

## Software and code

Policy information about availability of computer code

| Data collection | iBright1500 imaging system, SerialEM v4.0&v4.1, Thermo Scientific Xcalibur v4.1 |
|---|---|
| Data analysis | AlphaFold2, Amira v2020.2, AreTomo v1.3.4, Byonic by ProteinMetrics v5.1.1, Chimera v1.16 & ChimeraX v1.3&1.7, cryoSPARC v4.4, CTFFIND4, iBright1500 Analysis Software, IMOD v 4.11.3, ISOLDE v1.3, Motioncor2, Phenix v1.21, EMRinger v1.0.0, Prism v10.2.2 (GraphPad Software), Proteome Discoverer v2.1 (Thermo Fisher Scientific), PyCrESTA (https://github.com/brungerlab/pycresta), PyMol v2.3.2 (Schrödinger, LLC), Python v3.8, PySeg v1.0.0, PyTom v0.981a&1.1, R v3.4.2, RELION v3.1&4.0, Rosetta 2, Scientific Xcalibur (v4.1, Thermo Fisher Scientific), SerialEM v4.0&4.1, Situs v3.2, Topaz v0.2.5, Warp v1.0.9. Several script files are available at https://github.com/brungerlab/ISV_scripts. |

For manuscripts utilizing custom algorithms or software that are central to the research but not yet described in published literature, software must be made available to editors and reviewers. We strongly encourage code deposition in a community repository (e.g. GitHub). See the Nature Portfolio guidelines for submitting code & software for further information.

## Data

The subtomogram averaging maps (wild-type V0-only: 44858, wild-type State 1: 44855, wild-type State 2: 44856, wild-type State 3: 44857), the SPA maps (wild-type V0-only: 44846, wild-type State 1: 44843, wild-type State 2: 44839, wild-type State 3: 44840, Syp-/- V0-only: 44845, Syp-/- State 1: 44844, Syp-/- State 2: 44842, Syp-/- State 3: 44841) and representative binned tomograms (Syp-/- ISV:44847, wild-type ISV:44848) have been deposited in the EMDB, and the atomic coordinates have been deposited in the PDB (wild-type V0-only: 9BRZ, wild-type State 1: 9BRT, wild-type State 2: 9BRA, wild-type State 3: 9BRQ, Syp-/- V0-only: 9BRY, Syp-/- State 1: 9BRU, Syp-/- State 2: 9BRS, Syp-/- State 3: 9BRR).

## Research involving human participants, their data, or biological material

Policy information about studies with human participants or human data. See also policy information about sex, gender (identity/presentation), and sexual orientation and race, ethnicity and racism.

| Reporting on sex and gender | N/A |
|---|---|
| Reporting on race, ethnicity, or other socially relevant groupings | N/A |
| Population characteristics | N/A |
| Recruitment | N/A |
| Ethics oversight | N/A |

Note that full information on the approval of the study protocol must also be provided in the manuscript.

# Field-specific reporting

Please select the one below that is the best fit for your research. If you are not sure, read the appropriate sections before making your selection.

☒ Life sciences ☐ Behavioural & social sciences ☐ Ecological, evolutionary & environmental sciences

For a reference copy of the document with all sections, see nature.com/documents/nr-reporting-summary-flat.pdf

# Life sciences study design

All studies must disclose on these points even when the disclosure is negative.

| Sample size | Experiments described in this study were performed with at least 3-9 samples for each group, which is consistent with sample sizes commonly used in similar studies in the field. And the sample sizes provided adequate power to detect significant effects. |
|---|---|
| Data exclusions | Since it is difficult to determine the true number of ISVs without any V-ATPase assembly due to the missing wedge effect of tomographic reconstructions, we fitted the Poisson distributions to copy numbers ≥ 1. |
| Replication | All successful data were generated over multiple attempts and in at least three replicates (replicates number are specified in the manuscript) |
| Randomization | Experiments randomization was not necessary, but in all experiments, control and experimental samples were analyzed in parallel. |
| Blinding | Two individuals independently examined two halves of the tomograms and counted the copy numbers of V-ATPase per ISV. The two individuals knew the samples of the dataset, but didn't know each other's counting results. The mice observational behavior was scored blind by two individuals. |

# Reporting for specific materials, systems and methods

We require information from authors about some types of materials, experimental systems and methods used in many studies. Here, indicate whether each material, system or method listed is relevant to your study. If you are not sure if a list item applies to your research, read the appropriate section before selecting a response.

## Materials & experimental systems

| n/a | Involved in the study |
|-----|----------------------|
| ☐ | ☒ Antibodies |
| ☒ | ☐ Eukaryotic cell lines |
| ☒ | ☐ Palaeontology and archaeology |
| ☐ | ☒ Animals and other organisms |
| ☒ | ☐ Clinical data |
| ☒ | ☐ Dual use research of concern |
| ☒ | ☐ Plants |

## Methods

| n/a | Involved in the study |
|-----|----------------------|
| ☒ | ☐ ChIP-seq |
| ☒ | ☐ Flow cytometry |
| ☒ | ☐ MRI-based neuroimaging |

# Antibodies

| | |
|---|---|
| Antibodies used | mouse-VGLUT1(1:200, SySy, Cat.# 135303); rabbit-synaptophysin-1 (1:1,000; SySy, Cat.# 101008); rabbit-IRDye800CW (1:3,000; LI-COR, Cat.# 926-32211); mouse-synaptobrevin-2 (1:1,000; SySy, Cat.# 104211); mouse-HRP (1:10,000; Abcam, Cat.# ab6789); mouse-synaptotagmin-1 (1:1,000; SySy, Cat.# 105011); mouse-IRDye800CW (L1:3,000; I-COR, Cat.# 926-32210); rabbit-synaptoporin-1 (1:500; SySy, Cat.# 102002); rabbit-ATP6V1A1 (1:1,000; NovusBio, Cat.# NBP1-89342); rabbit-synaptogyrin-1 (1:500; SySy, Cat.# 103002); rabbit-VGLUT1 (1:1,000; SySy, Cat.# 135303); |
| Validation | mouse-VGLUT1 validated PubMed: 15103023; rabbit-synaptophysin-1 is validated by the company; mouse-synaptobrevin-2 validated PubMed: 26663078; mouse-synaptotagmin-1 validated PubMed: 29274147; rabbit-synaptoporin-1 validated PubMed: 31090538; rabbit-ATP6V1A1 validated of the orthogonal strategies by the company ; rabbit-synaptogyrin-1 validated PubMed: 31090538; rabbit-VGLUT1 validated PubMed: 15103023. |

# Animals and other research organisms

Policy information about studies involving animals; ARRIVE guidelines recommended for reporting animal research, and Sex and Gender in Research

| | |
|---|---|
| Laboratory animals | Male wild-type CD1 mice (23-26 days old) and C57BL6 mice (23-26 days old) were used for synaptic vesicle preparations. Male Syp -/- mice (23-26 days old) were used for synaptic vesicle preparations. Both sexes of 4-6 months-old wild-type Black 6 (B6NTac) and Syp-/- mice were used in animal behavior experiments. |
| Wild animals | N/A |
| Reporting on sex | Sexes were reported above |
| Field-collected samples | N/A |
| Ethics oversight | All animal procedures were performed in accordance with the National Institutes of Health Guide for the Care and Use of Laboratory Animals and approved by the Stanford Administrative Panel on Laboratory Animal Care (APLAC) institutional guidelines (protocol #29981) and by the University of Colorado Boulder Institutional Animal Care and Use Committee (IACUC) (protocol #1106.02). No field collected samples were used in the study. |

Note that full information on the approval of the study protocol must also be provided in the manuscript.

# Plants

| | |
|---|---|
| Seed stocks | N/A |
| Novel plant genotypes | N/A |
| Authentication | N/A |

