## [Peer Review file · Nature]

Manuscript Title: Structure and Topography of the Synaptic V-ATPase–Synaptophysin Complex

Reviewer Comments & Author Rebuttals

Reviewer Reports on the Initial Version:

Referee #1 (Remarks to the Author):

The authors analyze here synaptic vesicles, relying on cryo-electron microscopy and tomography, both in situ and in isolated synaptic vesicles. The most easily observed functional complex on the synaptic vesicle, the V-ATPase, is measured with relatively high efficiency. This molecule, as the authors point out, is responsible for establishing the pH value inside vesicles, and, indirectly, serves for neurotransmitter uptake. Its potential interaction with synaptophysin, a bona fide synaptic vesicle marker whose function is poorly understood, is studied by comparing the V-ATPase complex from wild-type and synaptophysin knock-out mice.

Overall, the authors show that:

- A molecular density, which corresponds reasonably well to synaptophysin, neighbors the V-ATPase in wild-type, but not synaptophysin knock-out mice
- The knock-out mice contain more V-ATPase copies per vesicle than normal mice, and have a higher propensity to seizures

The work is well performed, and shows convincing evidence, but I am unsure whether the level of novelty is sufficient for publication in Nature. A potential interaction between synaptophysin and the V-ATPase has been suggested almost 30 years ago, as the authors admit (their reference 32; Galli et al., JBC, 1996, doi: 10.1074/jbc.271.4.2193), and more recent cross-linking data also support this evidence. Therefore, the interaction suggested here is not particularly new.

Synaptophysin has long been proposed to participate in synaptic vesicle recycling, and to support the trafficking of other vesicle proteins, as synaptobrevin/VAMP (Gordon et al., J Neurosci, 2011, DOI:10.1523/JNEUROSCI.3162-11.2011). Therefore, while the potential involvement of synaptophysin in the recycling and/or packaging of the V-ATPase in vesicles is a new finding, it is not surprising, since similar functions have been suggested before.

Finally, a propensity to seizures for the knock-out mice is potentially interesting, but substantially more mechanistic data would be required before publishing the work solely on the merit of this observation.

Referee #2 (Remarks to the Author):

The manuscript by Wang et al., examines the interaction between the vesicular ATPase and the vesicular protein synaptophysin. Rather than looking at purified proteins in isolation, this study used cryo-EM with subtomogram averaging to examine the vesicular ATPase on purified glutamatergic synaptic vesicles isolated from mouse brain. With this approach, the authors achieved a resolution of ≈ 4 angstroms. A central premise of the study is the revelation of an extra density in the structure suggestive of a 20kDa protein. Leading candidates for this protein were determined using predictions provided by Alpha-Fold, with synaptophysin emerging as the most likely candidate. Importantly, prior structural studies of reconstituted V-ATPases failed to show this interaction, highlighting the critical importance of studying native synaptic vesicles. That synaptophysin interacts at this site was supported with the use of vesicles from synaptophysin KO mice, which revealed the loss of the extra density. Interestingly, in the *syp*^{-/-} mice, the number of V-ATPases per vesicle was increased. A slight weakness of this the study is that the *syp*^{-/-} mice were shown to exhibit an increase in seizure susceptibility, however no data were presented to explain why this might occur or to relate this susceptibility to the increase in V-ATPases/vesicle. It would be helpful if the authors could discuss this in a little more detail. That said, this is a cutting-edge structural study of the synaptic vesicle V-ATPase and its interaction with *syp*. It is well-presented. In addition, the extended data sets beautifully and convincingly support the main data sets.

Specific comments

1. Earlier work on *syp*^{-/-} mice did not reveal a strong phenotype with regards to changes in release probability or various forms of short or long term changes in synaptic strength. It is rather puzzling and interesting to see in the current study that there is an increase in seizure susceptibility. This begs the question of why the current results are more concordant with the quadruple KO than the *syn* KO alone (lines 251-252).
2. Can you definitively rule out a role for synaptogyrin or synaptoporin as interacting with the V-ATPase? Do you know whether these proteins are properly trafficked to the vesicles in the absence of *syp*?
3. With an increase in the intact V-ATPase number per vesicle in the absence of *syp*, a simple-minded expectation is that there would be an increase in quantal size. By contrast, an earlier report of *syp*^{-/-} mice suggested, that if anything, there was a modest decrease in glutamatergic quantal amplitude in the absence of *syp*. What happens to quantal size in the current study? What happens if one were to OE the ATPase without altering *syp* levels? Does this alone increase seizure phenotype, one wonders?
4. Please provide a more detailed explanation of what you mean by “vesicle biogenesis” (p. 11) and clarify the roles of synaptophysin and the V-ATPase/synaptophysin complex.

Referee #3 (Remarks to the Author):

Wang et al present a detailed study exploring the interaction between synaptic vesicle V-ATPase and synaptophysin using cryo-electron tomography and in situ single particle cryo-electron microscopy. The authors identify synaptophysin as a binding partner for V-ATPase and provide insights into the structural arrangements influenced by this interaction. Through knockout mouse models, they also demonstrate the physiological implications of the absence of synaptophysin, linking it to altered neurotransmitter release and seizure susceptibility.

The study is elegant, well-executed and provides significant novel insights into synaptic vesicle biology. The findings are compelling, particularly the link between synaptophysin and synaptic vesicle biogenesis, and the physiological consequences observed in knockout mice. These contribute valuable knowledge to the field of neurobiology.

However, some points need clarification or elaboration to strengthen the conclusions and the robustness of data interpretation.

Extended Fig 1. It is slightly concerning that classes 3 and 4, which were used for the final maps, represent only 6% of the particles, while class 1, which in cryoEM jargon would be considered as a junk class constitutes 86%. This distribution raises questions about the representativeness of the selected classes and whether they accurately reflect the majority of the particle population. Have the authors tried to further classify the particles belonging to class 1? Have they tried to classify the particles in more than 5 classes initially? Also it is not clear how the 1323 intact V-ATPase particles and the 573 V0 particles that were refined were selected. The workflow suggests that class 3 and 4 particles were used (i.e. about 60 particles) but there were obviously many more particles.

Page 4 line 107. "We note that the resolution of our V0-only map exceeds that of other purified mammalian V0-only V-ATPase structures (ref 27).".

Cite other more recent publications which are mentioned later in the manuscript. This reference is from 2001 and it is therefore not surprising that the additional density could not be seen.

page 5 line 115. "The top hit was synaptophysin (also called synaptophysin-1) 6, followed by its paralogs synaptoporin (also called synaptophysin-2) 7 and synaptogyrin-1 & -3"

This should be reformulated. Synaptoporin, and not synaptophysin, is the first of the 3 mentioned protein but it is "only" at position 8 of the correlation ranking. Synaptophysin is at position 11, synaptogyrin-1 at position 28, and synaptogyrin-3 at position 33.

page 6 line 147: "Finally, the extra density we observed in the wild-type ISVs was absent in Syp-/- ISV SPA maps for both intact and V0-only V-ATPase assemblies (Fig. 2g, h and Extended Data Fig. 4b-f), unequivocally validating the identity of the extra density as synaptophysin."

Unequivocally is too strong a statement. I agree that the conclusion is highly likely correct, but we could imagine that the absence of synaptophysin prevents other partners of V-ATPase such as synaptogyrin or synaptoporin to be integrated in SV. The authors actually report a decreased amount of synaptobrevin in the synaptophysin knock out and they mention in the discussion that "synaptophysin plays a role in synaptobrevin sorting into synaptic vesicles" (page 10, line 256).

Therefore the alternative interpretation that I mention may be plausible.

page 7 line 170 "However, it does not interact with the V-ATPase since there is no density for it in the Syp-/- ISV SPA maps".

It is not because we cannot see it that it does not interact.

The authors actually give this very explanation at page 10 line 267: "Since we did not observe densities for synaptobrevin or synaptophysin oligomers in our cryo-ET and SPA maps (Fig. 1c, d and Fig. 2a, b), it is likely that these interactions are heterogeneous and averaged out in the EM maps." The same can be said for synaptoporin or synaptogyrin.

page 6 line 161. "While both e1 (ATP6V0E1) and e2 (ATP6V0E2) isoforms have been reported for subunit e, our high-resolution V0-only V-ATPase map suggests e2 as the subunit present in our sample (Extended Data Fig. 6a)."

The very tightly cropped views of the map and model shown in the figure do not really allow a clear view. A bit more of density context would help the reader to assess the statement.

page 8 line 195. "Additionally, since it is difficult to determine the true copy number of ISVs without any V-ATPase assembly due to the missing wedge effect of tomographic reconstructions"

The same argument can be used for any copy number. It is clear that identifying V-ATPase in certain orientations is difficult due to the missing wedge. This results in a systematic underestimate of the copy number. This should be made clear and the authors could estimate the uncertainty in copy number due to the missing wedge.

Page 10, line 267. Reference 48 reports an interaction between V-ATPase and synaptobrevin, not between synaptophysin and synaptobrevin like suggested by the authors.

Page 15, line 385.

Instead of "pellet" should it read "incubated"?

Page 17, line 427. How was it stripped?

Page 21, line 543.

Why not use AI-based particle picker such as DeePiCt?

In extended data figure 1, typo: 1,323 articles  particles

Referee #4 (Remarks to the Author):

This manuscript reports a cryo-EM study of synaptic vesicles (SV) and demonstrate a stoichiometric association of synaptophysin with the proton pump. While the occurrence of a complex between synaptophysin, VAMP and the V0 section of the V-ATPase had been reported nearly 30 years ago, the occurrence of such a tight stoichiometric complex between the two proteins suggests a key physiological function. Surprisingly, while brains SV membranes contain 3 synaptophysin paralogues (synaptoporin, synaptogyrin 1 and 3), only synaptophysin associates with the V-ATPase. Based on studies of SVs purified from synaptophysin KO mice, the presence of synaptophysin did not appear to have an impact on the structure of the V-ATPase. However, the absence of synaptophysin increased the copy number of V-ATPases on SVs, up to 8 ATPases on a single SV, instead of only 1 or 2. The authors suggest that given the abundance of synaptophysin on SVs, its presence may limit the number of v-ATPases by molecular crowding.

Overall, this is a very good and solid study. Its main limitation is the lack of clear insight into the physiological significance of the synaptophysin-V-ATPase interaction. While this manuscript reports for the first time a significant functional defect in synaptophysin KO mice (an increase susceptibility to seizures), the mechanistic link between the synaptophysin-V0 interaction and seizures remains unclear. In spite of this limitation, I think this study will be of interest to a broad readership as it will spawn new interest in synaptophysin, a major, but still poorly understood, SV protein.

Can the authors comments on whether the interaction with the V-ATPase is mutually exclusive with the hexameric assembly of synaptophysin?

A manuscript reporting very similar findings has been recently posted in BioRxiv:
<https://doi.org/10.1101/2024.04.01.587493>

Clearly the two papers are synergistic. The authors should quote this other study.

Author Rebuttals to Initial Comments:

Below are the point-by-point responses to the reviewers' comments. The reviewer's comments are in *Italics*, and our responses are in blue. We have also provided a "related manuscript" file with revisions tracked.

Reviewer #1:

The authors analyze here synaptic vesicles, relying on cryo-electron microscopy and tomography, both in situ and in isolated synaptic vesicles. The most easily observed functional complex on the synaptic vesicle, the V-ATPase, is measured with relatively high efficiency. This molecule, as the authors point out, is responsible for establishing the pH value inside vesicles and, indirectly, serves for neurotransmitter uptake. Its potential interaction with synaptophysin, a bona fide synaptic vesicle marker whose function is poorly understood, is studied by comparing the V-ATPase complex from wild-type and synaptophysin knock-out mice.

Overall, the authors show that:

- *A molecular density, which corresponds reasonably well to synaptophysin, neighbors the V-ATPase in wild-type, but not synaptophysin knock-out mice*
- *The knock-out mice contain more V-ATPase copies per vesicle than normal mice, and have a higher propensity to seizures*

The work is well performed, and shows convincing evidence,

We thank the reviewer for the positive assessment of our work.

but I am unsure whether the level of novelty is sufficient for publication in Nature. A potential interaction between synaptophysin and the V-ATPase has been suggested almost 30 years ago, as the authors admit (their reference 32; Galli et al., JBC, 1996, doi: 10.1074/jbc.271.4.2193), and more recent cross-linking data also support this evidence. Therefore, the interaction suggested here is not particularly new.

Synaptophysin has long been proposed to participate in synaptic vesicle recycling, and to support the trafficking of other vesicle proteins, such as synaptobrevin/VAMP (Gordon et al., J Neurosci, 2011, DOI:10.1523/JNEUROSCI.3162-11.2011). Therefore, while the potential involvement of synaptophysin in the recycling and/or packaging of the V-ATPase in vesicles is a new finding, it is not surprising, since similar functions have been suggested before. Finally, a propensity to seizures for the knock-out mice is potentially interesting, but substantially more mechanistic data would be required before publishing the work solely on the merit of this observation.

Perhaps it is not surprising that synaptophysin and the V-ATPase interact, considering the suggestions made in the studies by Galli et al. (1996) and Wittig et al. (2021). What is striking and novel in our study is the dramatic impact of V-ATPase number during SV biogenesis and the resultant seizure susceptibility phenotype. These observations themselves imply the critical role of synaptophysin in the synaptic vesicle cycle far beyond those previously ascribed. Moreover, the molecular basis of the V-ATPase–synaptophysin interaction was previously unknown. Our structures, for the first time, provide an atomic-level description of this interaction, and our studies show that it is an interaction that is specific for synaptophysin but not synaptoporin and synaptogyrin. Our structures suggest that the synaptic vesicle membrane likely plays a major role in stabilizing this interaction, providing a unique insight into the importance of the membrane context for macromolecular interactions in general and, more specifically, for the biogenesis of synaptic vesicles.

While numerous pharmacological studies confirm that an imbalance between inhibitory and excitatory signals triggers seizures, despite decades of research, a mechanism that creates a persistent increase in the probability of spontaneous seizures has been lacking. Our observation that loss of synaptophysin results in an increase in V-ATPase number per synaptic vesicle provides a viable mechanistic explanation regarding seizure susceptibility, given that multiple studies have demonstrated seizure phenotypes in V-ATPase mutations and some of these mutations are close to the V-ATPase—synaptophysin interface (see also our response to Reviewer #4, below). Importantly, we observed that *Syp*^{-/-} mice exhibit a severe seizure phenotype under pharmacological treatment with a glutamate agonist (Fig. 4d), which is concordant with a small but significant increase in EPSC quantal amplitude (reference 40). More profound changes were observed with double genetic deletion of both synaptophysin and synaptogyrin, resulting in reduced short-term and long-term plasticity in neurons (reference 39). Quadruple knockout had an even more pronounced effect on quantal size (reference 43). This increased quantal size may lead to an excitatory to inhibitory imbalance under the stress of pharmacological treatment with a glutamate agonist.

A detailed mechanistic understanding of seizure susceptibility in *Syp*^{-/-} mice is a fascinating long-term goal. However, this would require a substantial effort far beyond the scope of this work, considering the general lack of knowledge regarding seizure mechanisms. Additionally, given that this is the first major phenotype observed for *Syp*^{-/-} mice, after almost 40 years of effort, untangling a detailed mechanistic explanation of how the loss of *Syp* and increased V-ATPase results in seizure susceptibility is also far beyond the scope of this work. We believe that our work is a major step in understanding SV biogenesis by providing the molecular details of V-ATPase—synaptophysin interaction and uncovering a mechanism for how this interaction is involved in regulating the V-ATPase copy number.

Referee #2 (Remarks to the Author):

*The manuscript by Wang et al., examines the interaction between the vesicular ATPase and the vesicular protein synaptophysin. Rather than looking at purified proteins in isolation, this study used cryo-EM with subtomogram averaging to examine the vesicular ATPase on purified glutamatergic synaptic vesicles isolated from mouse brain. With this approach, the authors achieved a resolution of ≈ 4 angstroms. A central premise of the study is the revelation of an extra density in the structure suggestive of a 20kDa protein. Leading candidates for this protein were determined using predictions provided by Alpha-Fold, with synaptophysin emerging as the most likely candidate. Importantly, prior structural studies of reconstituted V-ATPases failed to show this interaction, highlighting the critical importance of studying native synaptic vesicles. That synaptophysin interacts at this site was supported with the use of vesicles from synaptophysin KO mice, which revealed the loss of the extra density. Interestingly, in the *syp*^{-/-} mice, the number of V-ATPases per vesicle was increased. A slight weakness of this the study is that the *syp*^{-/-} mice were shown to exhibit an increase in seizure susceptibility, however no data were presented to explain why this might occur or to relate this susceptibility to the increase in V-ATPases/vesicle. It would be helpful if the authors could discuss this in a little more detail.*

We thank the reviewer for their positive assessment of our work. As we pointed out in our response to Reviewer 4, we would like to point out that multiple studies have described seizure phenotypes due to mutations in the V-ATPase (references 52-55), and our observation of substantially altered V-ATPase copy number per SV is a plausible contributor to the observed phenotype. We have added these references and discussion to the concluding paragraph.

As we mentioned in our response to Reviewer #1, we observed that *Syp*^{-/-} mice exhibit a severe seizure phenotype under pharmacological treatment (Fig. 4d), which is concordant with the increase in quantal size observed in knockout studies (references 40, 43). This increased quantal size may lead to an imbalance in neurotransmitter uptake and release.

*That said, this is a cutting-edge structural study of the synaptic vesicle V-ATPase and its interaction with *syp*. It is well-presented. In addition, the extended data sets beautifully and convincingly support the main data sets.*

We thank the reviewer for their very positive comments.

Specific comments

*1. Earlier work on *syp*^{-/-} mice did not reveal a strong phenotype with regards to changes in release probability or various forms of short or long term changes in synaptic strength. It is rather puzzling and interesting to see in the current study that there is an increase in seizure susceptibility. This begs the question of why the current results are more concordant with the quadruple KO than the *syn* KO alone (lines 251-252).*

See our response to point 3 below.

*2. Can you definitively rule out a role for synaptogyrin or synaptoporin as interacting with the V-ATPase? Do you know whether these proteins are properly trafficked to the vesicles in the absence of *syp*?*

We thank the reviewer for the comment. As we also point out in response to Reviewer #3 below, it is possible, albeit unlikely, that some molecules might interact weakly or dynamically with the V-ATPase in the *Syp*^{-/-} ISVs. Nevertheless, the complete absence of density in that region and the dramatic effect of V-ATPase copy number speak against the continuous and quantitative presence of binding partners in the *Syp*^{-/-} ISVs. Our western blots of *Syp*^{-/-} ISVs (Figure 2d and Extended Data

Figure 3) suggest that both synaptoporin and synaptogyrin are properly trafficked to the synaptic vesicles.

3. With an increase in the intact V-ATPase number per vesicle in the absence of syp, a simple-minded expectation is that there would be an increase in quantal size. By contrast, an earlier report of syp^{-/-} mice suggested, that if anything, there was a modest decrease in glutamatergic quantal amplitude in the absence of syp. What happens to quantal size in the current study? What happens if one were to OE the ATPase without altering syp levels? Does this alone increase seizure phenotype, one wonders?

The expectation that increased V-ATPase would increase quantal size is what is observed for both the QKO (reference 43 - Fig 2D) and the Syp^{-/-} mice (reference 40 - Fig 3D). This finding is consistent with the observed increase being attributed to synaptophysin. While the over-expression of an ATPase is an interesting experiment, it would be difficult to uniformly overexpress the multiple genes of the V-ATPase.

4. Please provide a more detailed explanation of what you mean by “vesicle biogenesis” (p. 11) and clarify the roles of synaptophysin and the V-ATPase/synaptophysin complex.

We suggest that the interaction between synaptophysin and the V-ATPase helps to establish the proper copy number for the V-ATPase. We changed the concluding sentence.

Reviewer #3:

Wang et al present a detailed study exploring the interaction between synaptic vesicle V-ATPase and synaptophysin using cryo-electron tomography and in situ single particle cryo-electron microscopy. The authors identify synaptophysin as a binding partner for V-ATPase and provide insights into the structural arrangements influenced by this interaction. Through knockout mouse models, they also demonstrate the physiological implications of the absence of synaptophysin, linking it to altered neurotransmitter release and seizure susceptibility.

The study is elegant, well-executed and provides significant novel insights into synaptic vesicle biology. The findings are compelling, particularly the link between synaptophysin and synaptic vesicle biogenesis, and the physiological consequences observed in knockout mice. These contribute valuable knowledge to the field of neurobiology.

REPLY: We thank the reviewer for their positive assessment of our work.

However, some points need clarification or elaboration to strengthen the conclusions and the robustness of data interpretation.

Point 1:

Extended Fig 1. It is slightly concerning that classes 3 and 4, which were used for the final maps, represent only 6% of the particles, while class 1, which in cryoEM jargon would be considered as a junk class constitutes 86%. This distribution raises questions about the representativeness of the selected classes and whether they accurately reflect the majority of the particle population. Have the authors tried to further classify the particles belonging to class 1? Have they tried to classify the particles in more than 5 classes initially? Also it is not clear how the 1323 intact V-ATPase particles and the 573 V0 particles that were refined were selected. The workflow suggests that class 3 and 4 particles were used (i.e. about 60 particles) but there were obviously many more particles.

REPLY:

We thank the reviewer for the comments. We apologize that the scheme in Extended Data Fig. 1 was unclear. We initially employed a template-free approach for V-ATPase identification, as described in the Methods section. In the first round of subtomogram averaging, only six tomograms were utilized. All putative membrane-bound densities were first extracted using PySeg. This generous density selection method explains the high percentage of non-V-ATPase particles in the “junk” class. Following several rounds of 2D and 3D classification, classes 3 and 4 displayed promising characteristics resembling the known structure of V-ATPase. These classes were subsequently combined and used as a template for an entirely new particle identification round (consisting of template matching and visual confirmation) using the entire dataset of 52 tomograms. This approach successfully identified 1,323 intact V-ATPase particles and 537 V0-subcomplex particles. To clarify our workflow, we have revised Extended Data Figure 1 and the corresponding Methods section.

Point 2:

Page 4 line 107. "We note that the resolution of our V0-only map exceeds that of other purified mammalian V0-only V-ATPase structures (ref 27).".

Cite other more recent publications which are mentioned later in the manuscript. This reference is from 2001 and it is therefore not surprising that the additional density could not be seen.

REPLY:

We double-checked the literature, and the only peer-reviewed structure of a mammalian V0-only V-ATPase assembly is the one cited (reference 27). Since we submitted our manuscript, we have become aware of two preprints posted on BioRxiv (<https://doi.org/10.1101/2024.04.01.587493> and <https://doi.org/10.1101/2024.04.11.588828>). Based on the information provided in the first preprint, the quality of that V0-only map appears to be comparable to that of our V0-only map.

Point 3:

page 5 line 115. "The top hit was synaptophysin (also called synaptophysin-1) 6, followed by its paralogs synaptoporin (also called synaptophysin-2) 7 and synaptogyrin-1 & -3" This should be reformulated. Synaptoporin, and not synaptophysin, is the first of the 3 mentioned protein but it is "only" at position 8 of the correlation ranking. Synaptophysin is at position 11, synaptogyrin-1 at position 28, and synaptogyrin-3 at position 33.

REPLY:

We thank the reviewer for the comment and apologize for the unclear description of our selection process. As mentioned in the Methods (Section on Unbiased Matching of the Extra Density to Protein Candidates), we visually checked the 200 top hits with reasonable CCS scores. Candidates that were not membrane proteins or had no membrane domain matching the observed extra density in the membrane region were ignored for further analysis. Among those that appeared reasonable, we adjusted the fit using ChimeraX and then calculated the percentage of outliers using ChimeraX. Synaptophysin had the lowest outlier percentage after adjustment, followed by synaptoporin and synaptogyrin-1 & -3 (Figure 2c). We have revised the description of our procedure in the main text and the corresponding Methods section.

Point 4:

page 6 line 147: "Finally, the extra density we observed in the wild-type ISVs was absent in Syp-/- ISV SPA maps for both intact and V0-only V-ATPase assemblies (Fig. 2g, h and Extended Data Fig. 4b-f), unequivocally validating the identity of the extra density as synaptophysin."

Unequivocally is too strong a statement. I agree that the conclusion is highly likely correct, but we could imagine that the absence of synaptophysin prevents other partners of V-ATPase such as synaptogyrin or synaptoporin to be integrated in SV. The authors actually report a decreased amount of synaptobrevin in the synaptophysin knock out and they mention in the discussion that "synaptophysin plays a role in synaptobrevin sorting into synaptic vesicles" (page 10, line 256). Therefore the alternative interpretation that I mention may be plausible.

REPLY:

We thank the reviewer for the comment. However, we kindly would like to point out that our western blots suggest that synaptoporin was at a somewhat higher level and synaptogyrin-1 was at a similar level as in the wild-type ISVs (Figure 2d and Extended Data Fig. 3). Thus,

there is no deficit of these two molecules in the Syp^{-/-} ISVs. However, we agree with the reviewer that “unequivocally” is too strong, so we have removed that adjective.

Point 5:

page 7 line 170 "However, it does not interact with the V-ATPase since there is no density for it in the Syp^{-/-} ISV SPA maps".

It is not because we cannot see it that it does not interact.

The authors actually give this very explanation at page 10 line 267: "Since we did not observe densities for synaptobrevin or synaptophysin oligomers in our cryo-ET and SPA maps (Fig. 1c, d and Fig. 2a, b), it is likely that these interactions are heterogeneous and averaged out in the EM maps."

The same can be said for synaptoporin or synaptogyrin.

REPLY:

We thank the reviewer for the comment. We agree that it is possible, albeit unlikely, that some molecules (in particular, synaptoporin) might interact weakly or dynamically with the V-ATPase in the Syp^{-/-} ISVs. Nevertheless, the complete absence of density in that region and the dramatic effect of V-ATPase copy number speak against the continuous and quantitative presence of binding partners in the Syp^{-/-} ISVs. We agree that the absence of density does not rule out the lack of weak or dynamic interactions. We have changed the sentence to

“However, it is improbable that it interacts quantitatively with the V-ATPase since there is no density for it in the Syp^{-/-} ISV SPA maps.”

Point 6:

page 6 line 161. "While both e1 (ATP6V0E1) and e2 (ATP6V0E2) isoforms have been reported for subunit e, our high-resolution V0-only V-ATPase map suggests e2 as the subunit present

in our sample (Extended Data Fig. 6a)."

The very tightly cropped views of the map and model shown in the figure do not really allow a clear view. A bit more of density context would help the reader to assess the statement.

REPLY:

We thank the reviewer for the comment. There are differences in 24 amino acid positions between e1 and e2. Out of these 24 positions, there are four positions where our density map allows discrimination between the corresponding e1 and e2 residues. We have provided a somewhat larger view of these regions for more context.

Point 7:

page 8 line 195. "Additionally, since it is difficult to determine the true copy number of ISVs without any V-ATPase assembly due to the missing wedge effect of tomographic reconstructions"

The same argument can be used for any copy number. It is clear that identifying V-ATPase in certain orientations is difficult due to the missing wedge. This results in a systematic underestimate of the copy number. This should be made clear and the authors could estimate the uncertainty in copy number due to the missing wedge.

REPLY

Indeed, the missing wedge affects the copy number of V-ATPase, as we mentioned in the main text (page 6, line 195). We, therefore, performed fits to Poisson distributions, but excluded the observed numbers of ISVs without observed V-ATPase assemblies from the fitting. Poisson distributions could be well fit to the copy numbers ≥ 1 (Extended Data Figs. 8e-h), suggesting that the incorporation of a V-ATPase is a Poisson process, *i.e.*, independent of the presence of other V-ATPases in the same synaptic vesicle. Focusing on the case of intact V-ATPase assemblies, the Poisson fits predict the number of ISVs without intact V-ATPases and suggest that our observations *overestimated* the number of ISVs without intact V-ATPases. This degree of overestimation is largely due to the missing wedge effect. On the other hand, the missing wedge effect also implies that we *underestimated* the number of intact V-ATPases for copy numbers ≥ 1 . However, the λ parameter of the fitted Poisson distribution should be independent of the missed fraction of V-ATPases, so it represents the true average number of intact V-ATPases per ISV. We have clarified these points in the main text and added more explanation to Methods.

In this context, we also would like to point out that to ensure consistent underestimation across both datasets, we employed identical data collection and processing parameters (including instruments, magnification, tilt range, dose, reconstruction, etc.). Additionally, two independent individuals performed the particle-picking process.

Point 8:

Page 10, line 267. Reference 48 reports an interaction between V-ATPase and synaptobrevin, not between synaptophysin and synaptobrevin like suggested by the authors.

REPLY:

We apologize for the confusion. We have replaced the reference with the proper reference.

Point 9:

Page 15, line 385.

Instead of "pellet" should it read "incubated"?

REPLY: Thank you for the comment. Indeed, we incubated the Dynabeads pellet/sediment with the LP2&VGLUT1 solution for another 2 hours at 4°C. We have updated the Methods.

Point 10:

Page 17, line 427. How was it stripped?

REPLY: The primary and secondary antibodies were stripped from the western blot membranes following the manufacturer's instruction (Thermo Fisher, Cat.# 62300). We incubated the probed blot in the 1X stripping buffer for 15 minutes and washed it 3 times for 5 minutes each in TBST washing buffer, then incubated the membrane in 5% BSA blocking buffer for 30 minutes until re-probing. We have updated the Methods.

Point 11:

Page 21, line 543.

Why not use AI-based particle picker such as DeePiCt?

REPLY:

We thank the reviewer for the comment. While several AI-based particle pickers, including DeePiCt, have been recently developed, template matching with manual validation remains the most used method in the field. The results from this approach often serve as the 'ground truth' for training AI-based tools. We employed a relatively low cross-correlation value to minimize false negatives during template matching, resulting in roughly four times more peaks for manual validation. We applied this procedure across all three datasets.

Point 12:

In extended data figure 1, typo: 1,323 articles  particles

REPLY:

We apologize for the typo and have corrected it.

Reviewer #4:

This manuscript reports a cryo-EM study of synaptic vesicles (SV) and demonstrate a stoichiometric association of synaptophysin with the proton pump. While the occurrence of a complex between synaptophysin, VAMP and the V0 section of the V-ATPase had been reported nearly 30 years ago, the occurrence of such a tight stoichiometric complex between the two proteins suggests a key physiological function. Surprisingly, while brains SV membranes contain 3 synaptophysin paralogues (synaptoporin, synaptogyrin 1 and 3), only synaptophysin associates with the V-ATPase. Based on studies of SVs purified from synaptophysin KO mice, the presence of synaptophysin did not appear to have an impact on the structure of the V-ATPase. However, the absence of synaptophysin increased the copy number of V-ATPases on SVs, up to 8 ATPases on a single SV, instead of only 1 or 2. The authors suggest that given the abundance of synaptophysin on SVs, its presence may limit the number of v-ATPases by molecular crowding.

Overall, this is a very good and solid study. Its main limitation is the lack of clear insight into the physiological significance of the synaptophysin-V-ATPase interaction. While this manuscript reports for the first time a significant functional defect in synaptophysin KO mice (an increase susceptibility to seizures), the mechanistic link between the synaptophysin-V0 interaction and seizures remains unclear. In spite of this limitation, I think this study will be of interest to a broad readership as it will spawn new interest in synaptophysin, a major, but still poorly understood, SV protein.

We thank the reviewer for their positive assessment of our work and its impact on the understanding of synaptophysin and the biogenesis of synaptic vesicles. While the link between the V-ATPase—synaptophysin interaction and the seizure susceptibility in the Syp-/- mice is admittedly conjecture, we would like to point out that multiple studies have described seizure phenotypes due to mutations in the V-ATPase and our observation of substantially altered V-ATPase copy number per SV is a plausible contributor to the observed phenotype. Several known human mutations of the V-ATPase a1 subunit are near the interface with synaptophysin:

Subunit	NCBI Accession	Reference	
ATP6V0a1	NP_001123493.1	52, Indrawinata et al.	
	S477P		S477
	R495W		L496
	A505P		A506
	N527D		N528
	G551E		G552
	R740Q		R741
	R804H		R805
ATP6V0c1	NP_001185498.1	54, Mattison et al.	
	A22T		A22
	G29S		G29
	P58A		P58
	G63A		G63
	V74F		V74
	A95T		A95
A95P	A95		

R119W	R119
F137L	F137
A138P	A138
A149T	A149
L150F	L150

In the above figure, all mutations in the a1 subunit near the interface with synaptophysin are colored red, all distal a1 mutations are colored purple, and all mutations in the c subunits are colored blue.

The phenotypic similarity between these mutations and the synaptophysin knockout, in conjunction with our data, suggest that increased seizure susceptibility has a similar etiology in both cases.

Point 1:

Can the authors comments on whether the interaction with the V-ATPase is mutually exclusive with the hexameric assembly of synaptophysin?

REPLY:

We superimposed a model of the synaptophysin hexamer (ref. 42) on the structure of the V-ATPase-synaptophysin complex by aligning one of the synaptophysin protomers of the hexamer with the complex. This produced a significant clash with the V-ATPase. By adjusting the lateral orientations of the synaptophysin molecules in the hexamer, we were able to generate a putative model that allows the synaptophysin molecule that interacts with the V-ATPase to also participate in a hexamer:

However, this hypothetical arrangement would require considerable deformation of the synaptophysin hexamer to fit into the curved membrane of the synaptic vesicle.

We have revised the paragraph starting with “The copy number of V-ATPases is ...”.

Point 2:

A manuscript reporting very similar findings has been recently posted in

BioRxiv: <https://doi.org/10.1101/2024.04.01.587493>

Clearly the two papers are synergistic. The authors should quote this other study.

REPLY:

We thank the reviewer for the comment. We have included a citation to this work in our revised manuscript. The quality of their V-ATPase—synaptophysin complex structures appears comparable to our work. However, we kindly note that this preprint includes no studies with the *Syp*^{-/-} ISVs. Thus, their binding partner validation is not strong. Moreover, going beyond reporting the structure of the complex, our study additionally shows the dramatic effect on V-ATPase copy number and the seizure phenotype in *Syp*^{-/-} mice, providing a biological context for the interaction between synaptophysin and the V-ATPase.

In addition to these revisions, we have made the following minor revisions upon internal review of our manuscript:

1. We removed the circles (marked 1, 2, and 3) in Figure 1a.
2. A cropping artifact was removed at the bottom right of Figure 4a - right.
3. Updated Fig. 3c and ED Figs. 6b-d based on final refined and deposited models.
4. Replaced ED 2a and ED 4a with Topaz-denoised images for better visualization.
5. Added a scatter plot of V-ATPase copy number vs. ISV diameter for WT and *Syp*^{-/-} ISVs (Extended Data Fig. 8).
6. In Figures 1b and 4, the image scales were different. In the revision, we selected views of the same size for both WT and *Syp*^{-/-}.